# Strong absorption and ultrafast localisation in NaBiS$_2$ nanocrystals with slow charge-carrier recombination

Yi-Teng Huang [1,14], Seán R. Kavanagh [2,3,4,14], Marcello Righetto [5], Marin Rusu [6], Igal Levine [7], Thomas Unold [6], Szymon J. Zelewski [1,8], Alexander J. Sneyd[1], Kaiwen Zhang[9], Linjie Dai [1], Andrew J. Britton[10], Junzhi Ye[1], Jaakko Julin [11], Mari Napari [12], Zhilong Zhang [1], James Xiao[1], Mikko Laitinen[13], Laura Torrente-Murciano [9], Samuel D. Stranks [1,9], Akshay Rao [1], Laura M. Herz [5,13], David O. Scanlon [2,4], Aron Walsh [3,4] & Robert L. Z. Hoye [3] ✉

I-V-VI$_2$ ternary chalcogenides are gaining attention as earth-abundant, nontoxic, and air-stable absorbers for photovoltaic applications. However, the semiconductors explored thus far have slowly-rising absorption onsets, and their charge-carrier transport is not well understood yet. Herein, we investigate cation-disordered NaBiS$_2$ nanocrystals, which have a steep absorption onset, with absorption coefficients reaching >10$^5$ cm$^{-1}$ just above its pseudo-direct bandgap of 1.4 eV. Surprisingly, we also observe an ultrafast (picosecond-time scale) photoconductivity decay and long-lived charge-carrier population persisting for over one microsecond in NaBiS$_2$ nanocrystals. These unusual features arise because of the localised, non-bonding S $p$ character of the upper valence band, which leads to a high density of electronic states at the band edges, ultrafast localisation of spatially-separated electrons and holes, as well as the slow decay of trapped holes. This work reveals the critical role of cation disorder in these systems on both absorption characteristics and charge-carrier kinetics.

Strong optical absorption and long charge-carrier lifetimes are two critical properties for efficient solar absorbers. Historically, efforts at developing novel absorbers have focussed on the former property, favouring materials with direct bandgaps allowing for strong optical transition. On the other hand, efforts at extending charge-carrier lifetimes have focussed primarily on developing careful (and often expensive) processing methods to minimise non-radiative recombination rates through lowering defect densities. The photovoltaic (PV)

[1]Cavendish Laboratory, University of Cambridge, JJ Thomson Ave, Cambridge CB3 0HE, UK. [2]Department of Chemistry, University College London, 20 Gordon Street, London WC1H 0AJ, UK. [3]Department of Materials, Imperial College London, Exhibition Road, London SW7 2AZ, UK. [4]Thomas Young Centre, University College London, Gower Street, London WC1E 6BT, UK. [5]Department of Physics, University of Oxford, Clarendon Laboratory, Parks Road, Oxford OX1 3PU, UK. [6]Struktur und Dynamik von Energiematerialien, Helmholtz-Zentrum Berlin für Materialien und Energie, 14109 Berlin, Germany. [7]Helmholtz-Zentrum Berlin für Materialien und Energie GmbH, Kekuléstraße 5, 12489 Berlin, Germany. [8]Department of Semiconductor Materials Engineering, Faculty of Fundamental Problems of Technology, Wrocław University of Science and Technology, Wybrzeże Wyspiańskiego 27, 50-370 Wrocław, Poland. [9]Department of Chemical Engineering and Biotechnology, University of Cambridge, Philippa Fawcett Drive, CB3 0AS Cambridge, UK. [10]School of Chemical and Process Engineering, University of Leeds, LS2 9JT Leeds, UK. [11]Department of Physics, University of Jyväskylä, P.O. Box 35, University of Jyväskylä, 40014 Jyväskylä, Finland. [12]School of Electronics and Computer Science, University of Southampton, Southampton SO17 1BJ, UK. [13]Institute for Advanced Study, Technical University of Munich, Lichtenbergstrasse 2a, D-85748 Garching, Germany. [14]These authors contributed equally: Yi-Teng Huang, Seán R. Kavanagh. ✉e-mail: r.hoye@imperial.ac.uk

application of lead-halide perovskites just over a decade ago has brought to the fore the concept of 'defect tolerance', in which the long charge-carrier lifetimes (>100 ns) of lead-halide perovskites are maintained despite the presence of significant defect concentrations ($10^{14}$–$10^{16}$ cm$^{-3}$). The origin of this feature is generally attributed to the fact that the dominant defects in perovskites have energy levels close to the band edges (i.e., shallow) and thus have lower recombination rates, while defects with deep levels tend to have higher formation energies and therefore lower concentrations[1–4]. The importance of defect tolerance to the performance of lead-halide perovskite photovoltaics has motivated efforts to find alternative classes of materials that mimic the exceptional optoelectronic properties of lead-halide perovskites (i.e., perovskite-inspired materials), but which can also overcome their toxicity and stability limitations[5–9]. These efforts have primarily focussed on metal-halide semiconductors, such as Cs$_3$Bi$_2$I$_9$, BiI$_3$, InI, Cs$_2$SnI$_6$, among many other examples (see ref. 9). However, the charge-carrier lifetimes achieved in these explored materials have mostly been in the 1–10 ns range, and many of them have indirect bandgaps, leading to low absorption coefficients on the order of $10^4$ cm$^{-1}$ near the band-edge[10–13].

I-V-VI$_2$ ternary chalcogenides are potential perovskite-inspired materials that have been gaining interest recently. This has been fuelled by the strong rises in power conversion efficiency of AgBiS$_2$ PV devices, which have now reached a certified value of 8.85%[14], the highest for any bismuth-based solar absorber reported thus far. Bismuth-based compounds are particularly important because these materials have demonstrated no evidence for toxicity[15], and Bi$^{3+}$ shares many electronic and chemical similarities to Pb$^{2+}$ that are believed to be conducive to defect tolerance[9]. Compared to halides, chalcogenides are generally less prone to oxidation or degradation in moist environments, which enables improved stability[16]. Indeed, AgBiS$_2$ and CuSbS$_2$ have been demonstrated to be very stable in ambient air[17,18]. A further advantage of this ABZ$_2$ materials family (where A is a monovalent cation, B is Sb$^{3+}$ or Bi$^{3+}$, and Z is a chalcogen) is the wide tunability in structural and optoelectronic properties. This includes the ability to achieve isotropic cubic phases, as opposed to the anisotropic, low-dimensional structures that many perovskite-inspired metal-halides form[9].

However, the absorption coefficient of AgBiS$_2$ rises only slowly from its optical bandgap of approximately 1 eV[14,17,19], which will limit the open-circuit voltage ($V_{OC}$) and short-circuit current density ($J_{SC}$) to below the Shockley-Queisser limit for thin films. Whilst this has been mitigated to a certain extent by inducing more homogeneous metal cation disorder through annealing, the absorption coefficient still does not reach >$10^5$ cm$^{-1}$ until >0.5 eV above the bandgap[14]. Similarly slowly-rising absorption onsets have been found in NaSbS$_2$, which (according to computational evaluation) does not reach $\alpha > 10^5$ cm$^{-1}$ until 1.7 eV above its bandgap, owing to parity-forbidden transitions at the bandgap energy[20]. Addressing these challenges to achieve strong optical absorption near the band edges would optimize the performance of these materials in ultrathin PV cells – a promising new frontier in the field of renewable energy research, which reduces the material demand (and thus the levelized cost of energy (LCOE)), aids flexible and non-obtrusive integration, as well as provides higher power-to-weight ratios than conventional thin-film PV (important for space applications)[21].

Beyond optical absorption, another important parameter that has not been discussed in the early-stage exploration of ABZ$_2$ materials is how charge-carriers couple to phonons, and how these interactions influence charge-carrier transport. Electronic coupling to longitudinal optical (LO) phonons in polar crystals leads to reductions in charge-carrier mobility owing to the formation of large polarons, whereas additional coupling to acoustic phonons can lead to carrier localisation and severe reductions in mobility through the formation of small polarons or self-trapped excitons[20,22]. Recent work on bismuth-halide

compounds has shown strong coupling between charge-carriers and phonons to be so common that it is coming to be regarded as a hallmark of these materials[23–27]. Understanding whether such effects occur in ABZ$_2$ systems will have significant implications on the future directions of exploration of this family of compounds.

In this work, we aim to address these critical questions through an in-depth investigation into the optoelectronic properties of NaBiS$_2$. This material is similar to AgBiS$_2$ in that the metal atoms can be disordered across the cation sublattice to give an effective high-symmetry cubic rocksalt phase (space group: $Fm\bar{3}m$)[28], due to the similar ionic radii of cations (129 pm for Ag$^+$, 116 pm for Na$^+$, 117 pm for Bi$^{3+}$)[29,30]. But unlike AgBiS$_2$, we would not expect the A-site cation in NaBiS$_2$ to contribute to the band-edge states since the filled 2$p$ orbitals of Na$^+$ are much further from vacuum level than the valence 6$s$ orbitals of Bi$^{3+}$ and 3$p$ orbitals of S$^{2-}$. Therefore the cation-anion hybridisation at the band edges of NaBiS$_2$ is expected to resemble that found between valence Pb$^{2+}$ and I$^-$ orbitals in lead-halide perovskites, with the A-site cation acting as a spectator ion[31]. The growth of phase-pure rocksalt NaBiS$_2$ nanocrystals (NCs) was recently achieved through nanocrystal synthesis[32], but the optoelectronic properties and carrier-phonon interactions of this material are not well established.

We firstly establish the absorption properties of NaBiS$_2$ NC thin films, and correlate them with density functional theory (DFT) calculations to understand the origin of the strong optical absorption. Next, to understand the kinetics of charge-carriers after photoexcitation, we use long-time transient absorption (TA) measurements, which are compared against measurements of photoconductivity decay using optical pump-terahertz probe (OPTP) spectroscopy to understand charge-carrier localisation behaviours. Short-time TA measurements, as well as calculations of the Fröhlich coupling constant and electronic density of states, are used to understand the nature of carrier-phonon coupling in this material. A model for describing the observed charge-carrier kinetics is also proposed. Finally, we investigate the impact of defects on charge-carrier kinetics in NaBiS$_2$ through post-annealing to introduce defects via ligand removal, which allows us to reveal the effect of such enhanced presence of traps with spectroscopic measurements.

## Results

### Synthesis, stability and absorption characteristics of NaBiS$_2$ nanocrystals

Adapting from the approach recently reported in ref. 32, we synthesized NaBiS$_2$ NCs from NaH, triphenyl bismuth and sulphur powder in a solution of oleylamine ligands (see details in Methods). Figure 1a shows the X-ray diffraction (XRD) patterns of NCs synthesised at 80 °C and 150 °C, respectively, which both match with the reference pattern for cation-disordered rocksalt NaBiS$_2$ (ICSD data base, Coll. Code: 616841). In this structure, both the Na$^+$ and Bi$^{3+}$ cations randomly occupy the same lattice sites, and are octahedrally coordinated by S$^{2-}$ anions (Fig. 1a, inset). Although the ordered trigonal phase ($R\bar{3}m$) is thermodynamically favoured at 0 K, the formation of the disordered rocksalt phase ($Fm\bar{3}m$) is entropically-driven at finite temperature and can be kinetically-stabilised via solution synthesis.

By adjusting the synthesis temperature, it was possible to tune the size of the synthesized NCs. Whilst we found the smaller NCs grown at 80 °C (5 ± 1 nm from transmission electron microscopy [TEM], Supplementary Fig. 1a, c and e) to be more colloidally-stable, the broad diffraction peaks were more difficult to distinguish. Therefore, to determine the phase-stability of the NaBiS$_2$ NCs in ambient air, we examined the larger NCs grown at 150 °C instead (18 ± 4 nm from TEM, Supplementary Fig. 1b, d and f). Figure 1b shows that both the XRD pattern and visual appearance of larger NaBiS$_2$ NCs did not change after 112 days at room temperature in ambient air, during which time the relative humidity varied between 60 and 70%. After 335 days in this environment, the same sample still remained phase-pure and

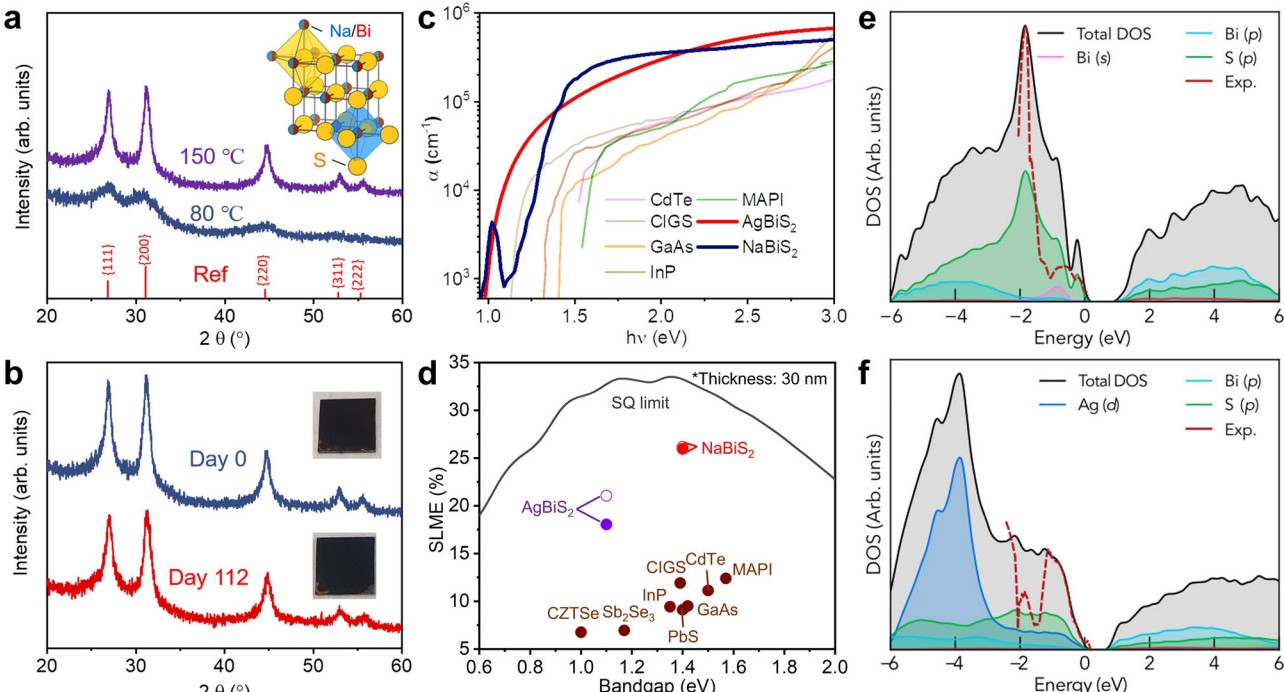

**Fig. 1 | Structural and optical properties of NaBiS$_2$ NC films. a** X-ray diffraction (XRD) patterns of NaBiS$_2$ nanocrystals (NCs) synthesized at 80 °C and 150 °C compared with the reference pattern for the disordered rocksalt ($Fm\bar{3}m$) phase of NaBiS$_2$[28]. **b** XRD patterns and photographs of NaBiS$_2$ NC films synthesized at 150 °C on the same day of preparation (Day 0) and after 112 days (Day 112) of storage in ambient air (60−70% relative humidity). **c** Absorption coefficient (α) spectrum of the NaBiS$_2$ NC film compared with other PV absorbers. **d** Spectroscopic Limited Maximum Efficiency (SLME) of various 30 nm-thick PV absorbers compared with the Shockley-Queisser limit (SQ limit, black line). The hollow and solid circles for NaBiS$_2$ and AgBiS$_2$ refer to the corresponding SLMEs without and with consideration of non-radiative losses owing to indirect bandgaps (details in Supplementary Note 1). The absorption coefficient spectrum used in (**c** and **d**) for the PV absorbers other than NaBiS$_2$ are extracted from refs. [14,74–80]. Theoretical and experimental (Exp.) orbital-projected electronic density of states (DOS) for disordered $Fm\bar{3}m$ **e** NaBiS$_2$ and **f** AgBiS$_2$. Theoretical DOS were calculated using hybrid DFT including spin-orbit interactions (HSE06+SOC) via the Special Quasirandom Structure (SQS) supercell approach. The energy of the highest occupied state is set to 0 eV. Experimental DOS were acquired from photoelectron yield spectroscopy (PYS) measurements on AgBiS$_2$ and NaBiS$_2$ NC films in the same experimental environment. The area under the partial DOS for atomic orbitals, as well as the total DOS are shaded for clarity.

maintained the same visual appearance, with only a slightly decrease in XRD peak intensity (Supplementary Fig. 2a). Similarly high air-stability is also expected in smaller NaBiS$_2$ NCs despite the difficulty to compare the relatively weak and broad XRD peaks (Supplementary Fig. 2b). We note that due to the higher NC colloidal stability, the remainder of the experiments in this work are based on drop-cast films of NaBiS$_2$ NCs synthesized at 80 °C.

To examine the absorption features in detail, we used ultraviolet-visible spectrophotometry (UV-Vis) to determine the absolute absorption coefficients (α), and photothermal deflection spectroscopy (PDS) to resolve the absorption profile more accurately at the band edge and below (details in Methods). Homogeneous films without any coffee-ring-like patterns (Fig. 1b, inset and Supplementary Fig. 3) were achieved by drop-casting the NC solution onto a glass substrate in an Ar-filled glovebox.

Surprisingly, our as-synthesised NaBiS$_2$ NCs show higher absorption coefficients than established direct-bandgap solar absorbers across a wide photon energy ($h\nu$) range from 1.3 eV to 3 eV, as shown in Fig. 1c. Unlike many other ABZ$_2$ materials[14,33–35], NaBiS$_2$ shows a steep absorption onset, reaching α > 10$^5$ cm$^{-1}$ just above its 1.4 eV bandgap (Fig. 1c; Tauc plot in Supplementary Fig. 4), which is characteristic of a direct optical transition. To illustrate the potential of NaBiS$_2$ for ultrathin PV, enabled by this extremely strong band-edge absorption, we calculated the Spectroscopic Limited Maximum Efficiency (SLME) for an ultrathin (30 nm) absorber layer. As shown in Fig. 1d, we obtained an 'ultrathin SLME' of 26% for NaBiS$_2$, while conventional thin-film PV absorbers only have values <12% at this thickness (further details in Supplementary Note 1 and Supplementary Table 1).

## Understanding the strong absorption of NaBiS$_2$ through first-principles calculations

Hybrid density functional theory (DFT) was employed to calculate the projected electronic density of states (DOS) of NaBiS$_2$ (Fig. 1e), compared with that of AgBiS$_2$ reported in ref. 14 (shown in Fig. 1f). In both materials, we find that the conduction band (CB) and lower valence band (VB) are both derived primarily from Bi $p$ and S $p$ states, demonstrating covalency and hence mixed ionic-covalent bonding in both materials[9,16,36].

However, significant differences are witnessed in the VB of both materials. In AgBiS$_2$, we see a large peak from Ag $d$ states at around 4 eV below the valence band maximum (VBM) with $p$-$d$ orbital repulsion from −3 eV up to the VBM. This results in a strong anti-bonding character at the VBM and significantly extends the VB bandwidth. In NaBiS$_2$ on the other hand, we find no such strong anti-bonding interaction, where Na$^+$ acts as a spectator ion with no orbital contribution to the VB DOS. Although an orbital repulsion between the occupied Bi 6$s^2$ lone-pair and S $p$ is found, the contribution is much weaker than Ag $d$ in AgBiS$_2$. As a result, NaBiS$_2$ shows a reduced VB bandwidth and a concentrated DOS near the VBM, with a distinct S $p$ peak around 2 eV below the VBM. In combination with a larger transition dipole moment, this concentrated DOS at the band edge results in extremely strong absorption in NaBiS$_2$. In order to directly compare the VB DOS spectrum of AgBiS$_2$ and NaBiS$_2$, the NC films based on both materials were measured through photoelectron yield spectroscopy (PYS) under the same experimental environment, in which we found the VB DOS of NaBiS$_2$ to reach larger values than AgBiS$_2$ at 2 eV below the VBM (dashed red lines in Fig. 1e and f).

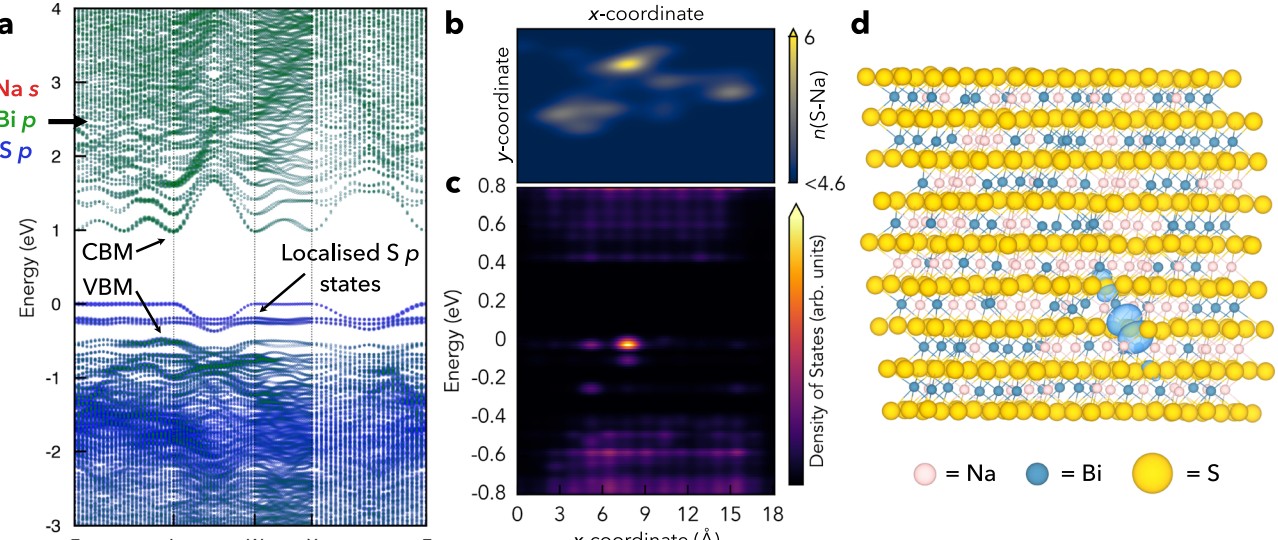

**Fig. 2 | Electronic structure and localised states in NaBiS₂. a** Orbital-projected effective electronic band structure of disordered $Fm\bar{3}m$ NaBiS₂ calculated using hybrid DFT including spin-orbit interactions (HSE06+SOC) via the SQS supercell approach. The supercell contains one S²⁻ anion coordinated by 6 Na⁺ cations, giving rise to the three localised S $p$ states above the delocalised valence band continuum. Sodium contributions (negligible) in red, bismuth in green and sulphur in blue. Corresponding Brillouin Zone path shown in Supplementary Fig. 6. **b** Heat map of the planar-averaged S−Na coordination ($n$(S−Na)) along the $xy$-plane of the NaBiS₂ SQS supercell. Yellow regions correspond to high coordination of S²⁻ anions with Na⁺ cations (octahedral coordination number ≥5). **c** Planar-averaged local electronic density of states (LDOS) of the NaBiS₂ SQS supercell around the bandgap. The colour bar is mapped to the normalised $\sqrt{\text{LDOS}}$. **d** Charge density isosurface (translucent blue) of the localised S $p$ states in the disordered NaBiS₂ SQS supercell. Sodium atoms in pink, bismuth in blue and sulphur in yellow. Isosurface set to 0.01 e Å⁻³.

Another consequence of the weak anti-bonding VBM character in disordered NaBiS₂, however, is the facilitated emergence of localised S $p$ states just above the 'bulk' or 'delocalised VBM' (Figs. 1e, 2a and Supplementary Fig. 5) at Na⁺-rich pockets (see next section for a detailed discussion). Here we refer to the states -0.5 eV below $E = 0$ eV in Fig. 1e as the 'bulk' VBM corresponding to typical semiconductor electron bands, whereas the localised S $p$ states above these bands occur only in low concentrations (-1 in 10¹⁷ unit cells) at local inhomogeneities in the cation distribution (Na⁺-rich pockets). Their prevalence thus depends on the cation distribution, and they behave akin to high concentration defects rather than band-like, delocalised electronic states.

These localised S $p$ states have a weaker transition dipole moment and relatively low concentrations compared with the CBM, thus do not significantly contribute to the absorption spectrum. Nevertheless, when plotted on a logarithmic scale, both the experimental (Fig. 1c) and calculated optical absorption (Supplementary Fig. 7a) show sub-gap peaks below the bandgap, in agreement with the presence of such localised states. To examine the existence of excitonic peaks as an alternative explanation, we fitted the absorption coefficient spectrum in Fig. 1c based on the Elliott model (see details in Supplementary Note 2). As a result, we obtained a small exciton binding energy of 12 meV, which is in good agreement with the calculated value of 27 meV for disordered NaBiS₂ using the Wannier-Mott hydrogenic model. Furthermore, as will be shown later in the Discussion section, excitons are less likely to form in disordered NaBiS₂ owing to the atomic-scale heterogeneity in the cation distribution and thus electronic potential. Therefore, we concluded that this sub-gap peak should not be excitonic.

The electronic band structure of disordered $Fm\bar{3}m$ NaBiS₂ is shown in Fig. 2a. Near the $L$ high-symmetry $k$-point, we find a disperse Bi $p$-derived conduction band minimum (CBM). On the other hand, we see a much less disperse 'bulk' VBM with only small Bi $s$ contributions, and the emergence of flat, localised S $p$ states just above this 'bulk' VBM. The weak dispersion of the VBM yields a pseudo-direct bandgap in disordered NaBiS₂, where the difference between the indirect

fundamental bandgap and the direct transition (Δ) is minimal (0.01 eV). Here, we obtain an indirect bandgap of 1.47 eV, with the 'bulk' VBM manifesting just off the $L$ point (in the $\Gamma$ direction), and the lowest-energy direct transition only 0.01 eV higher at the $L$ point. We found the estimated bandgap (1.4 eV) from the Tauc plot based on a direct-allowed transition (Supplementary Fig. 4) to be in good agreement with the calculated values. Also, it is worth mentioning that this small Δ is beneficial to achieving higher SLMEs for indirect-bandgap absorbers, since the corresponding reverse saturation current $J_0$ for them will not be substantially higher than the direct-gap case (see Supplementary Note 1). The SLME of NaBiS₂ when accounting for the non-radiative loss from a non-zero Δ only drops by 0.2% (Fig. 1d and Supplementary Table 1).

**Formation of localised S p states**

To further understand the role of cation-disorder on the formation of localised S $p$ states, we modelled the disordered structure using the Special Quasirandom Structure (SQS) approach, in which atoms in a simulation supercell are arranged to match the radial correlation functions of a truly random structure. By screening through a range of supercells, it was found that localised S $p$ states tend to form in Na⁺-rich regions, where ≥5 of the 6 octahedral coordinating neighbours of S²⁻ (Fig. 1a inset and Supplementary Fig. 5) are Na⁺ (rather than Bi³⁺). To illustrate, a planar-averaged heatmap of S−Na coordination (Fig. 2b) and the corresponding local electronic density of states (LDOS) (as a function of energy and position) within a representative supercell (Fig. 2c) are shown. Yellow patches in Fig. 2b signal Na⁺-rich regions with high S−Na coordination number (≥5), which correspond to high-energy spatially-localised electronic states – shown by the bright yellow region in Fig. 2c at the Fermi level (0 eV), contrasted with the delocalised CB and VB states -0.4 eV above and below. In fact, almost 70% of the highest occupied electronic state density in this supercell originates from the $p$ orbital of a single S atom (Fig. 2d). This behaviour was found to be consistent across various SQS supercells (from 80 to 400 atoms; see Methods), with local fluctuations of high Na⁺ density giving rise to high-energy localised S $p$ states

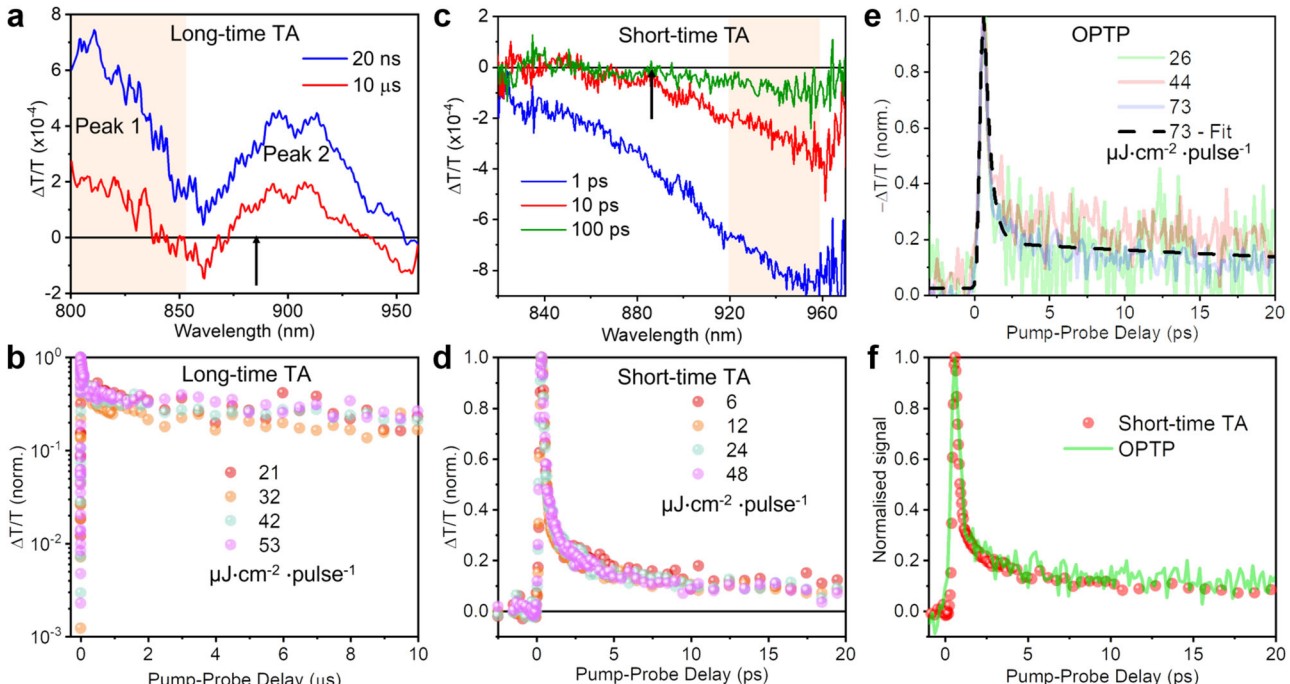

**Fig. 3 | Charge-carrier kinetics in NaBiS$_2$ NC films. a** Long-time transient absorption (TA) spectrum of NaBiS$_2$ NC films with 355 nm wavelength pump excitations and shown for pump-probe delays of 20 ns (blue) and 10 μs (red), along with **b** its normalised ground state bleach (GSB) signal kinetics at different pump fluences: 21 (dark pink), 32 (orange), 42 (green) and 53 (light pink) μJ cm$^{-2}$ pulse$^{-1}$. The GSB kinetics were acquired by averaging the signals from 800 to 860 nm (peak 1, shaded area in **a**) and normalised to the maximum $\triangle T/T$ values. **c** Short-time TA spectrum of NaBiS$_2$ NC films with 400 nm wavelength pump excitations at pump-probe delays of 1 ps (blue), 10 ps (red) and 100 ps (green), along with **d** its normalised photo-induced absorption (PIA) signal kinetics at different pump fluences: 6 (dark pink), 12 (orange), 24 (green) and 48 (light pink) μJ cm$^{-2}$ pulse$^{-1}$. The PIA kinetics were acquired by averaging the signals from 920 to 960 nm wavelength (shaded area in **c**) and normalised to the minimum $\triangle T/T$ values. The black arrows in (**a** and **c**) indicate the corresponding wavelength (886 nm) for the optical bandgap (1.4 eV) of NaBiS$_2$ (Supplementary Fig. 4). **e** Normalised $-\triangle T/T$ kinetics of spin-coated NaBiS$_2$ NC films from optical-pump-terahertz-probe (OPTP) measurements at different 400 nm wavelength pump fluences: 26 (green), 44 (pink) and 73 (purple) μJ cm$^{-2}$ pulse$^{-1}$. Black dashed line represents the fit to the kinetics measured at the highest fluence based on the two-level mobility model (Supplementary Note 4). **f** Comparison of the normalised signal kinetics acquired from the short-time TA (red circles) and OPTP measurements (green line), recorded at fluences of 12 and 26 μJ cm$^{-2}$ pulse$^{-1}$, respectively.

just above the 'delocalised VBM'. We note that similar formation of localised anion $p$ states at regions of low electronic potential, namely clusters of low-valence (A$^{I/II}$) cations, have recently been reported in the related A$^{II}$B$^{IV}$N$_2$ disordered compounds (including MgSnN$_2$, ZnSnN$_2$, ZnGeN$_2$, and others)[37–39], as well as disordered kesterites (CZTS)[40].

Using ambient-pressure X-ray photoemission spectroscopy (XPS), we found the as-synthesised NaBiS$_2$ NCs to have a relatively Na$^+$-rich surface (Na:Bi:S = 0.4:0.3:0.3, which is also Bi$^{3+}$-rich but S$^{2-}$-poor) in agreement with the observation of A$^+$-rich surfaces in AgBiS$_2$ NCs[17,41]. While ligand interactions could play a role, the highly Na$^+$-rich environment implies a higher concentration of such localised S $p$ states at the NC surface, where fast hole trapping is expected. We also note that all elements were present throughout the bulk of the material as well, as measured by Time-of-Flight Elastic Recoil Detector Analysis (ToF ERDA, see details in Supplementary Fig. 8).

## Charge-carrier kinetics in NaBiS$_2$ nanocrystal films

To understand the carrier kinetics of NaBiS$_2$ NCs, long- and short-time transient absorption (TA), as well as optical pump-terahertz probe (OPTP) measurements were performed on NaBiS$_2$ NC films.

In long-time TA measurements, we excited the film with 355 nm pump laser pulses of 0.8 ns duration, and used probe pulses comprising a broadband spectrum in the near-IR region to monitor the change in transmittance ($\triangle T/T$) of the NaBiS$_2$ film at certain delays after pump excitation (pump-probe delays), from 1 ns up to 100 μs. The positive ground state bleach (GSB) signal in a TA spectrum is usually proportional to the hole population near the VBM and

electron population near the CBM. The GSB signal decay can hence reflect the depopulation processes of charge-carriers near the band edges. In Fig. 3a, we observed two GSB signals peaking at 810 nm and 900 nm (peaks 1 and 2, respectively), which are both slightly off the wavelength corresponding to the estimated optical bandgap (1.4 eV or 886 nm, indicated by the black arrow in Fig. 3a), implying that the GSB signals here may not directly involve optical transitions at the band edges. The kinetics of both peaks could be acquired by averaging the signal intensity within 800–860 nm (shaded area in Fig. 3a) and 860–960 nm. We found that both peaks decay very slowly and follow almost the same kinetic behaviour (Supplementary Fig. 9), indicating that both peaks originate from the same photophysical species. Due to the higher intensity of peak 1, we monitored its kinetics under different fluences (21–53 μJ cm$^{-2}$ pulse$^{-1}$) and found it to be fluence-independent within this range (Fig. 3b), which suggests that the de-population process may not be significantly associated with bimolecular recombination or trap-filling effects[42,43]. Alternatively, it is also possible that the defect concentration in the NaBiS$_2$ NC film was much higher than the photogenerated charge-carrier density under the fluences used here so that the non-radiative recombination rate was hardly changed. However, we will show later in the Discussion section that the GSB kinetics are almost independent of the defect density within the NaBiS$_2$ NC film. It is surprising that this GSB signal is extremely long-lived and can still retain 20% of the early-time intensity measured within the first 800 ps resolution window after 5 μs. For comparison, among perovskite-inspired materials, Cs$_2$AgBiBr$_6$ has demonstrated some of the longest charge-carrier lifetimes, but its GSB signal can only retain 10% of its

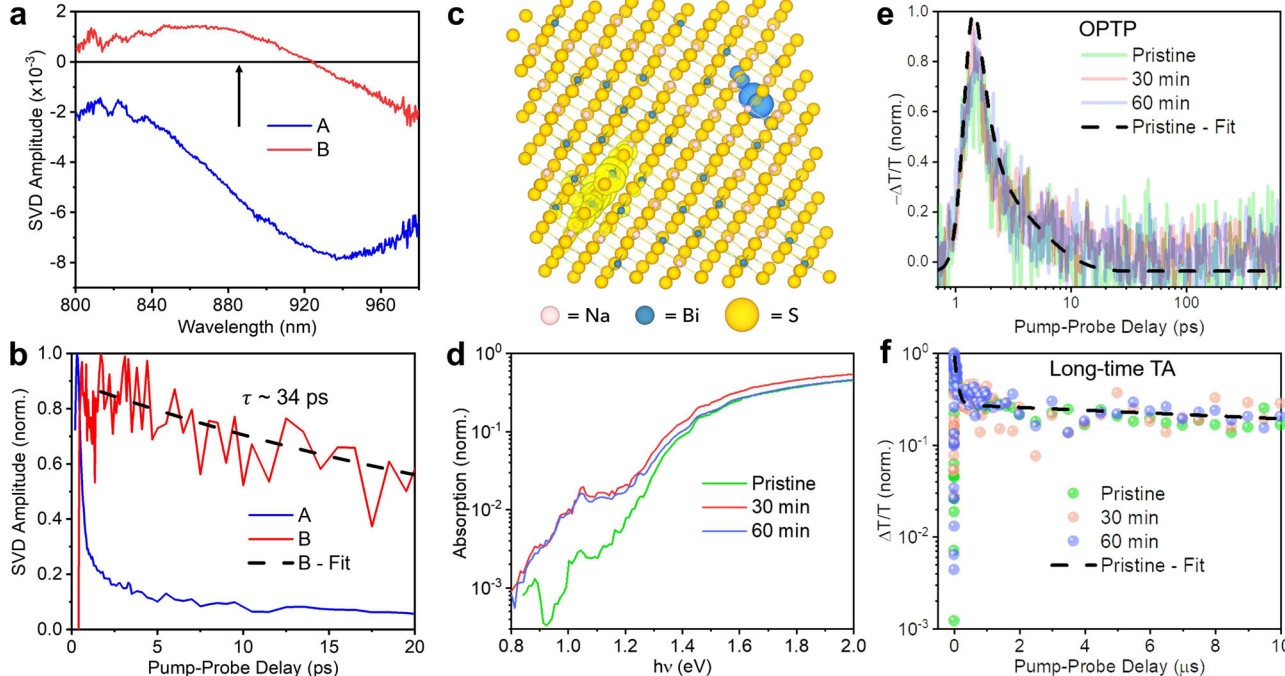

**Fig. 4 | Physical mechanism behind the unusual charge-carrier kinetics in NaBiS₂. a** Amplitude of the two principal components (A – blue, and B – red) extracted from the singular-value decomposition (SVD) analysis of the short-time TA spectrum. Black arrow indicates the wavelength corresponding to the optical bandgap of NaBiS₂. **b** Normalised kinetics of the two principal SVD components (colour legend same as in **a**), along with a monoexponential fit to component B (black dashed line). **c** Charge density isosurfaces of a relaxed electron-hole pair in a disordered NaBiS₂ supercell with sodium, bismuth, and sulphur atoms in pink, blue and yellow respectively. The translucent yellow and blue surfaces represent electron and hole densities, respectively. **d** Normalised absorption spectra of pristine and annealed (30 and 60 min) NaBiS₂ NC films. The annealing temperature was 100 °C. Normalised signal kinetics of pristine and annealed (30 and 60 min) NaBiS₂ NC films acquired from (**e**) OPTP and (**f**) long-time TA measurements. The OPTP measurements had 400 nm wavelength pump excitations at a fluence of 23 μJ cm⁻² pulse⁻¹, and the long-time TA measurements had 355 nm wavelength pump excitations at a fluence of 32 μJ cm⁻² pulse⁻¹. In (**d**–**f**), the pristine sample is represented in green, 30 min annealed sample in red, and 60 min annealed sample in purple lines or circles. The black dashed lines in (**e**) and (**f**) represent the fitted curve for the pristine film.

early-time intensity after 2 μs[9,44]. This emphasises the surprisingly slow nature of charge-carrier decay in NaBiS₂.

In short-time TA measurements, NaBiS₂ NC films were excited by 400 nm pump pulses of 100 fs duration, and similarly probed with broadband near-IR probe pulses to monitor the $\triangle T/T$ at shorter pump-probe delays from 1 ps to 1.7 ns. In contrast to the long-time TA results, we only observed a single negative photo-induced absorption (PIA) signal with a valley centred at 955 nm in the short-time TA spectrum (Fig. 3c), and so the signal intensity of 920–960 nm was averaged to extract the PIA kinetics (shaded area in Fig. 3c). Different from GSB signals, PIA signals typically result from charge-carrier transitions between intra-band states, or between trap/self-trapped states and the CB/VB. Figure 3d displays the PIA kinetics at varying pump fluences (6–48 μJ cm⁻² pulse⁻¹), and again a fluence-independent decay was witnessed. Unlike the long-lived GSB signal, the PIA signal intensity drops by around 80% within the first 1.5 ps after excitation, before subsequently decaying at a lower rate. We note that the PIA signal almost completely decays after 100 ps and no GSB signal emerges even at the upper limit of the measurable pump-probe delay (1.7 ns), which indicates that the zero pump-probe delay in long-time TA measurements might actually differ by at least ~2 ns. The origin of the ultrafast PIA decay but late emergence of GSB will be discussed later in the Discussion section.

Singular-value decomposition (SVD) analysis shows that the short-time TA spectrum is composed of two principal components, which are the dominant A and weaker B components shown in Fig. 4a. As discussed in Supplementary Note 3, component B is very likely to be associated with free charge-carrier bleaching near the band edges. We note that component B decays completely within 100 ps with a lifetime

of ~34 ps (Fig. 4b), suggesting that the long-lived GSB signals previously observed (Fig. 3a) are not directly caused by the de-population of free charge-carriers.

In order to understand the free charge-carrier kinetics, OPTP measurements were also performed on NaBiS₂ NC films. In OPTP measurements, similar 400 nm pump pulses (but of 35 fs duration) and a similar pump-probe delay window to short-time TA were employed, while terahertz (THz) radiation pulses were used to probe the free charge-carrier kinetics instead. As explained in Supplementary Note 4, the $-\triangle T/T$ signals in OPTP measurements are proportional to the photoconductivity $\triangle \sigma$ of the samples, and the signal kinetics therefore gives access to free charge-carrier dynamics as well as corresponding mobility. Owing to the limit on the maximum laser output power, spin-coated NaBiS₂ NC films with higher photoconductivity than drop-cast films were prepared here (see details in Methods) in order to investigate the $-\triangle T/T$ kinetics over a range of fluences. We observed an ultrafast $-\triangle T/T$ decay corresponding to an 80% drop in photoconductivity within the first 1–2 ps. Again, the signal kinetics were fluence-independent, as shown in Fig. 3e, which rules out the possibility of contributions from bimolecular recombination to the observed decay. We note that similar ultrafast and fluence-independent dynamics in OPTP transients have previously been reported for other bismuth-halide materials, such as Cs₂AgBiBr₆[26] and Cu₂AgBiI₆[27], in which intrinsic self-trapping has been regarded as the origin of ultrafast decay in $-\triangle T/T$ signals. Therefore, a similar self-trapping process may be also present in NaBiS₂.

By fitting the OPTP kinetics using the two-level mobility model previously developed for Cs₂AgBiBr₆[26] (see details in Supplementary Note 4), a close fit was obtained, as shown in Fig. 3e. We found the

effective electron-hole sum mobility (lower limit) to reduce from an initial value of $0.29\,cm^2\,V^{-1}\,s^{-1}$ immediately after excitation, to approximately an order of magnitude lower at $0.03\,cm^2\,V^{-1}\,s^{-1}$ after only 1.5 ps (Supplementary Table 2). The corresponding localisation rate in NaBiS$_2$ ($2.5 \pm 0.6\,ps^{-1}$) is faster than that reported in Cs$_2$AgBiBr$_6$ ($\sim1\,ps^{-1}$)[25,26], in which previous work has shown that a high deformation potential leads to self-trapping through strong coupling to acoustic phonons[25].

To determine the role of charge-carrier-phonon coupling in charge-carrier trapping, we first estimated the coupling strength in disordered NaBiS$_2$ by calculating the Fröhlich coupling constant $\alpha_{OP}$ given by:

$$\alpha_{OP} = \frac{1}{2}\frac{1}{4\pi\epsilon_0}\left(\frac{1}{\epsilon_{optical}} - \frac{1}{\epsilon_{static}}\right)\frac{e^2}{\hbar}\sqrt{\frac{2m_{eff}}{\hbar\omega_{LO}}} \qquad (1)$$

where $\epsilon_0$ is the vacuum permittivity, while $\epsilon_{optical}$ and $\epsilon_{static}$ are the calculated values of the dielectric function at high- (optical) and low-frequency (static). $\hbar$, $\omega_{LO}$, and $m_{eff}$ are the reduced Planck constant, effective longitudinal optical (LO) phonon frequency (calculated as $\omega_{LO} = 4.06$ THz; details in Methods), and charge-carrier effective mass, respectively. The static value of the dielectric function in NaBiS$_2$ is large ($\epsilon_{static} = 43.7$), due to the presence of highly polarisable Bi$^{3+}$ cation[9,16,36]. From Fig. 2a, it can be seen that the weak anti-bonding character within the VB of disordered NaBiS$_2$ leads to a less disperse VBM and hence a hole effective mass of $m_h = 1.04\,m_0$ that is twice as heavy as that of AgBiS$_2$ ($m_h = 0.51\,m_0$), although the electron effective masses for both materials are similarly small ($m_e = 0.24\,m_0$), owing to their shared Bi $p$ – S $p$ derived CBM. The large dielectric constant and heavier effective masses in disordered NaBiS$_2$ thus yield intermediate Fröhlich electron- and hole-phonon coupling constants ($\alpha_{OP}^e$ and $\alpha_{OP}^h$) of 1.40 and 2.92, which are higher than AgBiS$_2$ ($\alpha_{OP}^e = 1.09$, $\alpha_{OP}^h = 1.63$) and comparable to double perovskites as well as methylammonium lead iodide perovskites (2–3)[25,45]. The hole-phonon coupling constant here was calculated using the hole effective mass in the 'bulk VBM' rather than the localised S $p$ states arising at Na$^+$-rich inhomogeneities, owing to the breakdown of the effective mass model for such highly-localised states. In these localised states, hole coupling will be far stronger, as demonstrated by explicit polaron trapping calculations shown in the Discussion section. Therefore, our calculations of intermediate charge-carrier-phonon coupling strength for the 'bulk' band edges reflects the predisposition of NaBiS$_2$ to charge-carrier localisation, and the highly-localised S $p$ states can further enhance the strong hole-phonon coupling.

We note that exciton formation and defect trapping could also lead to an ultrafast decay in OPTP measurements. However, as discussed earlier, we believe that exciton formation is less likely in NaBiS$_2$ because of the: (i) low exciton binding energy (12 meV from Elliott model fitting; 27 meV from calculations), (ii) heterogeneity in the cation distribution favouring spatial electron-hole separation (see Discussion section), and (iii) OPTP kinetics being fluence-independent, which is consistent with one-step small polaron formation, rather than bimolecular exciton formation. In terms of defect trapping, we will also show in the Discussion section later that the overall charge-carrier kinetics of NaBiS$_2$ are mostly unchanged by an increase in defect density, confirming the presence of an intrinsic self-trapping/localisation process for charge-carriers.

## Discussion

### Mechanism for charge-carrier localisation

The presence of localised S $p$ states is associated with ultrafast charge-carrier localisation in NaBiS$_2$. From explicit calculations of the disordered NaBiS$_2$ supercells, we find that holes are preferentially trapped at the localised S $p$ states occurring at the Na$^+$-rich clusters (Fig. 4c), as expected. These clusters have significantly smaller S-Na bond lengths

(2.7–2.8 Å) than the average S-Na bond length (2.94 Å), which expand to ~2.87 Å upon hole trapping. The self-trapping (binding) energy of these small hole polarons is calculated to be ~0.2 eV (see Methods), with the exact value varying with the local environment (i.e., Na/Bi distribution) about the S$^{2-}$ anion. Moreover, we note that no initial structural perturbation was required to induce self-trapping (as is often required in simulations of polaronic trapping[46,47]), indicating a negligible energy barrier to hole trapping and thus rapid charge-carrier localisation[25]. This behaviour is consistent with low electronic dimensionality in semiconductors[45,48], and has been linked to similar ultrafast localisation in other bismuth-based materials[22,26]. While the close-packing of NaBiS$_2$ bestows high structural dimensionality, the effective electronic dimensionality of the band-edge states is vastly reduced due to the spectator nature of Na$^+$ and nanoscale heterogeneity in cation distribution. This leads to localised 0D S $p$ states in the highest-occupied (hole) electron bands (Na$^+$-rich pockets), and the lowest-energy unoccupied (electron) states primarily inhabiting Bi$^{3+}$-rich regions in the material. These computational results show that small but significant concentrations of localised S $p$ states at Na$^+$-rich pockets in NaBiS$_2$ can rapidly and strongly trap holes to form small polarons.

On the other hand, our calculations reveal that electron polarons are only weakly-localised in Bi$^{3+}$-rich regions (Fig. 4c), but this localisation process can be strengthened by a heterogeneous electronic potential due to atomic-scale fluctuations in the cation distribution (giving local Na$^+$-rich and Bi$^{3+}$-rich regions, as measured from ambient-pressure XPS). Our supercell calculations of electron-hole pairs in disordered NaBiS$_2$ reveal spontaneous separation of the excited charge-carriers into isolated polarons, rather than the formation of bound excitons. Thus, whilst excitons can form between the CBM and continua in the VB, we do not expect excitons to form between the CBM and localised S $p$ states. This localisation of electron and hole polarons at Bi$^{3+}$-rich and Na$^+$-rich pockets, respectively, results in a spatial segregation of the excited charge-carriers (Fig. 4c), which likely inhibits the formation of self-trapped excitons, and accounts for the absence of photoluminescence in NaBiS$_2$. We note that Na vacancies were also investigated as a potential origin of the sub-gap absorption and fast trapping in NaBiS$_2$ (details in Supplementary Note 5). However, we found them to be shallow acceptors. The exception was when these vacancies were located in Na$^+$-rich pockets, which then gave rise to deep traps. However, the concentration of these deep states would be too low to account for fast charge-carrier trapping because of their higher formation energies than elsewhere in the material, and the fact that they can only form in the Na$^+$-rich pockets, which already have low concentrations. Furthermore, the (0/−1) transition levels are so deep they are closer to the CBM than VBM, which would also not facilitate fast hole trapping, and is inconsistent with the energy of the sub-gap absorption peak relative to the optical bandgap found experimentally (Fig. 1c).

Interestingly, we found that the kinetics of both OPTP and short-time TA signals were almost identical (Fig. 3f), suggesting that the same charge-carriers were probed in both measurements. Considering that the OPTP signals are likely dominated by electrons owing to their much smaller effective mass and therefore higher mobility compared to holes in NaBiS$_2$, the PIA signals here are also likely to originate from the electron transition from the CB or self-trapped states to the higher excited state, similar to what has been reported in CdS and CdSe quantum dots[49]. Moreover, both the OPTP and PIA results suggest that an ultrafast localisation process should occur in the free electron population of NaBiS$_2$. We note here that although electrons have smaller Fröhlich coupling constant and weaker binding (to Bi$^{3+}$-rich regions) compared to holes, their localisation rate is still fast (since the OPTP signal is dominated by electrons and it decays fast), possibly owing to the significant enhancement from cation inhomogeneity.

Based on our measurements and calculations, we propose the following mechanism to describe the unusual charge-carrier kinetics in NaBiS$_2$ NC films. After photoexcitation, electrons are rapidly localised to the Bi$^{3+}$-rich regions owing to cation inhomogeneity, and holes are also rapidly localised at Na$^+$-rich clusters (localised S $p$ states), forming small polarons. However, because of the low dispersion of the VBM and thus low mobility, it may take a relatively long time for all photoexcited holes to reach those localised S $p$ states. The above processes account for the ultrafast decay in OPTP/short-time TA transients and slow emergence of long-lived GSB signals in long-time TA. A small fraction of residual free charge-carriers at band edges could relax mono-exponentially with a lifetime of ~34 ps via Shockley-Read-Hall recombination, as implied in Fig. 4b. Finally, owing to the spatial separation of trapped electrons and holes, radiative recombination of electrons and holes is very unlikely to occur in disordered NaBiS$_2$. Therefore, trapped electrons might relax non-radiatively to the ground state, possibly via a thermally-assisted process[50], which accounts for the slower decay part of the OPTP kinetics. On the other hand, self-trapped holes in the localised states can relax much more slowly, owing to the stabilisation from the strong polaron binding energy, which impedes non-radiative recombination and thus leads to the extremely slow decay of the broadband GSB signals in long-time TA measurements. We note that similar results would be expected in the case of bulk disordered NaBiS$_2$ films, though likely with differences in the localisation rates, due to the potentially weaker geometric confinement[51], as well as the effects of greatly-reduced surface area, or different synthesis conditions on the degree of cation heterogeneity.

## Influence of defects

In this section, we discuss the potential role of defects on the charge-carrier kinetics of NaBiS$_2$ NC films. It has been shown that post-annealing films can cause the intentional introduction of defects[52], or tune the homogeneity of cation disorder[14]. The latter, in the case of NaBiS$_2$, would influence the degree of localised S $p$ state formation. We therefore annealed NaBiS$_2$ NC films at different temperatures (50, 100, and 150 °C) for 1 hour in an Ar-filled glovebox. We found that all post-annealed films remained in the cation-disordered rocksalt phase. But in the case of films annealed 150 °C, we observed a slight shift in XRD peaks to higher diffraction angles (Supplementary Fig. 10a). The reduced lattice constants associated with these peak shifts may be due to improved cation homogeneity, which is predicted to lead to reduced cation-anion bond lengths (Supplementary Fig. 11), consistent with previous reports of post-annealed AgBiS$_2$[14]. But unlike AgBiS$_2$, we found post-annealed NaBiS$_2$ to exhibit decreased absorbance in the above-gap region and increased absorbance in the sub-gap region (Supplementary Fig. 10b). Further increasing the annealing temperature of NaBiS$_2$ to higher values resulted in the degradation of the material to orthorhombic Bi$_2$S$_3$ from 250 °C (Supplementary Fig. 12). Thus, over the limited range of post-annealing temperatures available, NaBiS$_2$ remains in the disordered phase, and we are limited in the extent to which cation homogeneity could be improved, such that localised states cannot be eliminated.

The increased absorbance at photon energies below 1.3 eV implies that sub-gap trap states may be introduced via annealing. Whilst NaBiS$_2$ is a cation-disordered solid, these NCs are crystalline with well-defined cation and anion sites in a rocksalt crystal structure. Thus, point defects (e.g., cation/anion vacancies, interstitials, cation on anion and anion on cation anti-sites) can still occur. Although large defect concentrations are very unlikely to form at such low annealing temperatures, oleic acid or oleylamine ligands surrounding the NaBiS$_2$ NCs could be detached during heating and remove surface species such as Bi atoms, which may create dangling bonds and thus defect states on the NC surface. To verify this postulation, we compared the change in sub-gap absorption of films composed of small NCs (mean size ~5 nm, determined from TEM images) versus films composed of

large NCs (mean size ~18 nm, determined from TEM images) after the same post-annealing treatment. If defects are mainly introduced to the NC surface after annealing, we would expect to observe a smaller change in sub-gap absorption for larger NCs owing to their lower surface area-to-volume ratio. We indeed found this to be the case, as shown in Supplementary Fig. 13. Furthermore, although a significant Bi stoichiometric change was observed in the ambient-pressure XPS spectrum (Supplementary Fig. 10c), we did not see a clear trend in the change in the bulk composition of the annealed samples (Supplementary Fig. 8), which again suggests that the introduction of defects can occur more easily on the surface than in the bulk of the NCs. Also, annealed NaBiS$_2$ films showed a more inhomogeneous morphology with several voids, which indicates that NCs could have fused together after ligand removal, as shown in Supplementary Fig. 14. This NC fusion process can also introduce sub-gap trap states by forming 'necked dimers', as previously found in PbS quantum dots[53]. We therefore conclude that the post-annealing treatment introduces sub-gap defects into the NaBiS$_2$ NC films, predominantly to the surface of the NCs. It is worth mentioning that the absorption intensity of the sub-gap peak at ~1 eV also increases in the annealed samples, further excluding the excitonic character of this sub-gap peak since the absorption of an excitonic peak should not depend on defect concentration.

Interestingly, even with the introduction of additional sub-gap defects, the charge-carrier kinetics in OPTP, short-time and long-time TA for drop-cast NaBiS$_2$ NC films remain almost the same (Fig. 4e, f, and Supplementary Fig. 15). For the samples annealed at 100 °C for different times (30 and 60 min), the OPTP dynamics can still be well described by the same two-level mobility model used in Fig. 3e, and the delocalised mobility $\mu_{del}$ as well as localisation rate $k_{loc}$ have been extracted for comparison (see details in Supplementary Note 4). As shown in Supplementary Table 2, we can see that $\mu_{del}$ has been increased from ~0.14 cm$^2$ V$^{-1}$ s$^{-1}$ in the pristine film to ~0.27 cm$^2$ V$^{-1}$ s$^{-1}$ in the film annealed for 60 min, which suggests that charge-carrier transport might be improved as a result of NC fusion after annealing. We note here that $\mu_{del}$ for the pristine film is also slightly lower than that for the spin-coated film measured in Fig. 3c, which can be attributed to the closer packing of the NCs within the spin-coated film as a result of ligand-exchange treatment. However, we do not see a clear trend in $k_{loc}$, which indicates that the ultrafast localisation in NaBiS$_2$ does not strongly depend on defect concentration. This provides further confirmation that the rapid decay in the OPTP kinetics is caused by self-trapping rather than defect trapping, since an increase in defect density should otherwise increase the trapping rate. In addition, GSB kinetics from long-time TA measurements could also be fitted by a phenomenological bi-exponential model $A_1 e^{-(t-t_0)/\tau_1} + A_2 e^{-(t-t_0)/\tau_2}$ with $t$ being the pump-probe delay, $t_0$ the pump-probe delay at which the maximum GSB occurs after photoexcitation, $\tau_1$ and $\tau_2$ the fitted time constants, and $A_1$ and $A_2$ the fitted pre-exponential constants. From this fitted model, an effective lifetime $\tau_{eff} = \frac{A_1 \tau_1 + A_2 \tau_2}{A_1 + A_2}$ can be numerically estimated. We note here there is no significant change in the fitted curves and constants (Fig. 4f and Supplementary Table 3) for the annealed films, which indicates that defects are also not greatly involved in the slow decay kinetics. All of the above results demonstrate that both ultrafast charge-carrier localisation and slow relaxation of trapped holes in disordered NaBiS$_2$ are not significantly influenced by the introduction of defects. Instead, they are mainly dominated by intrinsic atomic-scale cation heterogeneities.

In conclusion, we have found NaBiS$_2$ to have higher absorption coefficients than established direct-bandgap thin film absorbers, as well as an absorption onset steeper than AgBiS$_2$ and other ABZ$_2$ materials. These absorption properties arise owing to the high DOS in the upper VB, as well as the pseudo-direct nature of the bandgap. However, the Na$^+$ spectator character also leads to the formation of localised S $p$ states above the VBM, which is accentuated by an inhomogeneous

$Na^+$-$Bi^{3+}$ cation distribution, such that localised states emerge at $Na^+$-rich clusters. These localised S $p$ states cause strong hole self-trapping and results in a slow relaxation process exceeding several microseconds, which are not influenced by intentionally introduced trap states through post-annealing treatment. Although charge-carrier localisation leads to a reduction in sum mobility by almost an order of magnitude within a few picoseconds, this drawback may yet be mitigated by the strong absorption and long-lived photogenerated charge-carriers, which might open up the possibility of using these materials in ultrathin solar cells. More broadly, our work shows that the chemistry of the elemental species used in $ABZ_2$ materials enables strong control over the optical and transport properties through the electronic structure, and that charge-carrier-phonon coupling is a critical factor that needs to be accounted for in the future design of $ABZ_2$ materials.

## Methods

### NaBiS₂ NC synthesis

7.2 mg NaH (dry, 90%, Merck), 132 mg triphenyl bismuth (99%, Alfa Aesar) and 32 mg sulphur powder (99.5%, Alfa Aesar) were dissolved in 10 mL degassed oleylamine (70%, Merck) under stirring at room temperature for 15 min. The solution was heated at 80 or 150 °C for 30 min after which the solution colour turned from red to black. All the above processes were performed in a glovebox. Later, the whole solution was cooled down to room temperature in a water bath, and mixed with 6 mL hexane (>95%, Merck) and 14 mL oleic acid (90%, Merck) for at least 2 h to replace most oleylamine ligands by strongly attached oleic acid ligands. Finally, acetone (anhydrous, >99.9%, ROMIL) and acetonitrile (anhydrous, >99.9%, ROMIL) was used to precipitate the synthesized NaBiS₂ NCs, and the purified NCs were re-dissolved in hexane.

### Optical measurements

The ultraviolet-visible spectrophotometry (UV-vis) absorption spectra were measured in a Shimadzu UV 3600 spectrometer equipped with an integrating sphere. For PDS measurements, NaBiS₂ films were drop-casted on Spectrosil® 2000 quartz substrates and immersed in an inert liquid FC-72 Fluorinert (3 M Company), which has a high thermo-optic coefficient. A monochromatic beam from a 100 W Xenon arc source (Photon Technology International) integrated with a 250 mm focal length monochromator (CVI DK240) was illuminated perpendicularly to the sample surface, modulated with a mechanical chopper at a frequency of 13 Hz. Non-radiative recombination processes at the film surface lead to a temperature gradient, and thus a refractive index gradient in the liquid surrounding the sample. A 670 nm CW diode laser beam passing through the immersive medium, parallel to the sample surface (transverse configuration) is deflected and detected by a quadrant photodiode, with the signal amplitude demodulated with a lock-in amplifier (Stanford Research Systems SR830).

For long-time TA measurements, the third harmonic (355 nm) of an electronically controlled, Q-switched Nd:YVO4 laser (Innolas Picolo 25) provided ~800 ps pump pulses. For short-time TA measurements, the second harmonic (400 nm) of the Ti:Sapphire laser provided ~100 fs pump pulses. Broad-band near-IR probe pulses ranging from 800 to 980 nm were provided by a noncolinear optical parametric amplifier (NOPA) setup. Probe pulses were split into two beams by a beamsplitter. The other reference beam can then be used to calibrate shot-to-shot noise coming from the NOPA setup itself. This allows very weak signals to be measured. Both the probe and reference beams were detected by a Si dual-line array detector read out by a custom-built board from Stresing Entwicklungsbüro. The transmittance with and without pump excitation ($T_{pump\,on}$ and $T_{pump\,off}$) were collected alternatively at a repetition rate of 500 Hz, and the TA signals can be expressed as $\frac{\triangle T}{T} = \frac{T_{pump\,on} - T_{pump\,off}}{T_{pump\,off}}$.

OPTP measurements were conducted by using a setup described in detail elsewhere[27]. Briefly, an amplified Ti:sapphire laser system (Spectra-Physics, Spitfire) provides 800 nm wavelength pulses of 35 fs

pulse duration and 5 kHz repetition rate. Single-cycle THz radiation pulses were generated via the inverse spin Hall effect upon photo-excitation of a spintronic emitter with the fundamental laser output[54]. THz detection was achieved by using a fraction of the fundamental laser output to gate the THz signal by free-space electro-optic (EO) sampling with a 1 mm-thick ZnTe (110) crystal. Here, a Wollaston prism was used to separate different circularly polarized components of the gate, which were then measured by a pair of balanced photodiodes. Samples were excited by frequency-doubled 400 nm pulses, obtained by second-harmonic generation in beta-barium-borate (BBO) crystal. During the OPTP measurements, the THz emitter, EO crystal, and samples are kept under vacuum at pressures below $10^{-1}$ mbar. For fluence-dependent measurements (Fig. 3c), samples were prepared by spin-coating the NaBiS₂ NC solution onto 2 mm thick z-cut quartz substrates, and 50 μL of 0.1 M NaI solution in methanol was then dropped onto the spin-coated layer for 2 min to perform ligand-exchange treatment. The film after ligand-exchange treatment was rinsed by methanol then hexane to remove the residual organic ligands. The long and insulating organic ligands would be replaced by shorter iodide-based ligands, which improves the photoconductivity within these spin-coated samples. In the annealing effect study (Fig. 4e), samples were prepared by drop-casting the NC solution onto 2 mm thick z-cut quartz substrates.

### Absorption measurements

The absorption coefficient α was calculated from Eq. 2 below:

$$\alpha = \frac{\ln\left(\frac{1-R}{T}\right)}{d} \tag{2}$$

where $R$ and $T$ are the reflectance and transmittance, respectively, of drop-cast NaBiS₂ NC films, and $d$ is the film thickness. $R$ and $T$ were measured by UV-Vis within an integrating sphere, and $d$ was determined from the cross-sectional profile of the step-edge across from a substrate to a film using an atomic force microscope (Nanoscope III), as shown in Supplementary Fig. 3. To ensure homogeneous films were prepared for these measurements, we drop-cast the NC solution onto a glass substrate in an Ar-filled glovebox. The hexane solvent used for the NC solution evaporated rapidly, and it can be seen from Fig. 1b that this resulted in uniform films with no coffee-ring patterns. Atomic force microscopy measurements showed that variations in film thickness were only on the order of 10 nm (Supplementary Fig. 3). The relative absorbance spectrum measured by PDS was normalised to its highest signal value. We then mapped this maximum value of the absorbance, found at 3.1 eV photon energy, onto the absolute absorption coefficient value obtained from UV-Vis at the same photon energy. The whole absolute absorption coefficient spectrum could then be acquired.

### DOS measurement

The PYS setup uses a probe to detect the photoemission currents as a function of incident photon energy. The light source comprises a deuterium ($D_2$) lamp coupled with a grating monochromator. The range of the incident photon energy is 3.4–7.5 eV. The sample is illuminated via a DUV optical fiber. The ionization energy of samples was determined by measuring the photoelectron yield $Y(h\nu)$ as a function of photon energy $h\nu$. $Y(h\nu)$ is defined as the number of photoemitted electrons per incident photon at a given photon energy $h\nu$. By extrapolating the linear part of the $Y^{1/3}(h\nu)$–$h\nu$ plot to $x$-axis, the ionization energy is found according to[55]:

$$Y(h\nu) \propto (h\nu - E_i)^3 \tag{3}$$

The photoemission threshold is determined with a resolution of 30 meV. In addition to the ionization energy, the PYS spectra contain

the information on effective DOS spreading from the Fermi level ($E_F$) down to $h\nu - \Phi$ below $E_F$[56]. The DOS spectra were obtained from the $Y(h\nu) - h\nu$ plots as the first derivative of the photoelectron yield $Y(h\nu)$ with respect to photon energy ($h\nu$), thus as $\frac{dY}{dh\nu}$.

With the known $\Phi$ and $E_i$, the energy level of VBM ($E_{VBM}$) is calculated as:

$$E_{VBM} = \Phi - E_i \tag{4}$$

Further details on the PYS setup and the applied methodology of evaluating the experimental data can be found elsewhere[55].

## X-ray characterization

XRD measurement was performed on a Bruker D8 Advance diffractometer. A copper $K_\alpha$ X-ray source ($\lambda = 1.5406$ Å) was used. Near ambient-pressure XPS measurements were conducted in an enviroESCA electron spectroscopy made by SPECS, which is equipped with a near ambient pressure Phoibos 150 analyser with one-dimensional delay line detectors. A monochromated Al $K_\alpha$ X-ray source ($\lambda = 8.3386$ Å) was utilised. All XPS measurements were conducted in an atmosphere of 7 mbar of Ar gas. A fast (<10 s) pump down to vacuum (<1 × $10^{-5}$ mbar) to remove residual air was performed before the venting to 7 mbar with Ar gas. The pass energies for the XPS survey and high-resolution measurement were 100 eV and 50 eV, respectively.

## TEM characterization

TEM samples were prepared by dropping the diluted $NaBiS_2$ NC solution onto a carbon-coated copper grid. TEM images were recorded in a FEI Tecnai F20 (120 kV) microscope.

## Theoretical methods

All calculations were performed using DFT within periodic boundary conditions through the Vienna Ab-initio Simulation Package[57–59]. Scalar-relativistic projector augmented-wave (PAW) potentials were used to describe the interaction between the core and valence electrons[60]. A plane-wave kinetic energy cutoff of 350 eV and Γ-centred k-point meshes with reciprocal space sampling of 0.38 Å$^{-1}$ were found to give energies converged to <1 meV atom$^{-1}$ for disordered supercells, and so were used for all calculations except for the electronic DOS, for which a denser sampling of 0.13 Å$^{-1}$ was used. Geometry relaxations were iterated until cell volumes were unchanged, to avoid Pulay stress.

To simulate the $Fm\bar{3}m$ disordered rocksalt crystal structure of $NaBiS_2$, the SQS approach was used[61], whereby supercells are generated with the cation–cation pair correlations optimized to match that of the ideal infinitely-random distribution. The Alloy Theoretic Automated Toolkit[62] was used to generate SQS supercells via Monte Carlo–simulated annealing[63]. Thirty Monte Carlo simulations were performed for each supercell size, with the structure giving the best match ('objective function') to a fully-random material chosen for further calculations.

The screened hybrid DFT exchange–correlation functional of Heyd, Scuseria and Ernzerhof (HSE06[64]) was used for the calculation of all structural and electronic properties, save for that of the ionic dielectric response, being well-established for the accurate description of semiconductor properties[65]. The ionic dielectric screening and phonon frequencies were calculated under Density Functional Perturbation Theory (DFPT) using semi-local DFT (PBEsol), due to the prohibitive cost of hybrid DFT with large supercells for these calculations and the established accuracy of PBEsol for this property[66], while the optical response was calculated using the method of Furthmüller et al. to obtain the high-frequency real and imaginary dielectric functions[67]. The effective LO phonon frequency for the calculation of Fröhlich carrier coupling was extracted from the weighted sum over dot products of phonon eigenvectors and dipole moments, averaged over the unit sphere, as implemented in the amset package[68]. Due to

the presence of the heavy-atom element Bi, spin–orbit-coupling (SOC) effects were included in all electronic and optical calculations.

Supercell sizes up to 400 atoms ($28 \times 27 \times 13$ Å$^3$) were calculated to ensure convergence in the energetic, electronic, optical and polaronic properties. The 400-atom supercell was used for the generation of LDOS plots and charge density isosurfaces, while a well-converged 80 atom supercell ($8 \times 17 \times 14$ Å$^3$) was used for calculation of the unfolded bandstructure due to a prohibitive computational cost and data storage requirement (>1 Tb). Electron and hole polaron calculations were performed by adding/removing an electron to/from the SQS simulation supercells, while 'excitonic' supercells were generated by constraining the total spin to give a triplet state – in each case using 80, 160 and 400 atoms to confirm supercell-independence of the results. A range of initial perturbations to the initial structure were tested using the Bond Distortion Method[47], to aid polaron trapping, though in each case the unperturbed structure also relaxed to the localised self-trapped state (indicating negligible trapping barriers). Polaron trapping/binding energies are taken from total energy differences of the hole or electron-containing supercells before and after relaxation of the atomic coordinates.

Unfolded electronic band structures and density of states were generated using *easyunfold*[69] and *sumo* respectively[70]. *Effmass* was used to calculate the carrier effective masses[71], and *pymatgen* was used throughout for analysis of calculation data[72]. Na vacancies were investigated by separately placing vacancies at each Na site in the SQS supercell, applying the ShakeNBreak defect structure searching approach[73] and calculating their charge-dependent formation energies (details in Supplementary Note 5).

## Data availability

The experimental and computational data generated in this paper and in the Supplementary Information have been deposited to the Research Data Repository at Imperial College London under the https://doi.org/10.14469/hpc/10614.

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

## Acknowledgements

The authors would like to thank Laura Spies and Yanhao Wang for their contributions in the early-stage development of $NaBiS_2$ NCs, and Sachin R. Rondiya, Yuchen Fu, and Kavya Reddy Dudipala for helpful discussions. Y.-T.H. would like to thank funding from the Ministry of Education, Taiwan as well as Downing College Cambridge. S.R.K. acknowledges the Engineering and Physical Sciences Research Council (EPSRC) Centre for Doctoral Training in the Advanced Characterisation of Materials (CDT-ACM) (no. EP/S023259/1) for funding a PhD studentship, as well as the UCL Kathleen High Performance Computing Facility (Kathleen@UCL), the Imperial College Research Computing Service and associated support services. By the membership of the UK's HEC Materials Chemistry Consortium, which is funded by EPSRC (no. EP/L000202, EP/R029431 and EP/T022213), this work used the ARCHER2 UK National Supercomputing Service and the UK Materials and Molecular Modelling (MMM) Hub (Young; EPSRC no. EP/T022213). L.M.H. and M.R. thank EPSRC for funding (no. EP/V010840/1). L.M.H. acknowledges support through a Hans Fischer Senior Fellowship from the Technical University of Munich's Institute for Advanced Study, funded by the German Excellence Initiative. S.J.Z. acknowledges support from the Polish National Agency for Academic Exchange within the Bekker programme (grant no. PPN/BEK/2020/1/00264/U/00001). I.L. acknowledges the AiF project (ZIM-KK5085302DF0) for financial support. A.J.S would like to thank the Royal Society Te Apārangi and the Cambridge Commonwealth European and International Trust for their financial support. A.J.B. acknowledges support from the Henry Royce Institute (EPSRC grants: EP/P022464/1, EP/R00661X/1), which funded the VXSF Facilities within the Bragg Centre for Materials Research at Leeds (https://engineering.leeds.ac.uk/vxsf). The ToF-ERDA measurements and analysis were supported by the RADIATE project under the Grant Agreement 824096 from the EU Research and Innovation programme HORIZON 2020. K.Z. would like to acknowledge the EPSRC Centre for Doctoral Training in Graphene Technology (no. EP/L016087/1) for studentship. D.O.S. acknowledges support from EPSRC (no. EP/N01572X/1) and the European Research Council, ERC (no. 758345). S.D.S. acknowledges support from the Royal Society and Tata Group (no. UF150033), EPSRC (no. EP/R023980/1 and EP/S030638/1) and the European Research Council (ERC) under the European Union's Horizon 2020 research and innovation programme (HYPERION, no. 756962). A.R. acknowledges support from EPSRC. R.L.Z.H. would like to thank the Royal Academy of Engineering through the Research Fellowship scheme (no. RF\201718\1701) and EPSRC (no. EP/V014498/1).

## Author contributions

R.L.Z.H. and Y.-T.H. conceived of this project and developed the process for growing $NaBiS_2$ nanocrystals and films. Y.-T.H. synthesised NCs, performed UV-Vis, TA and XRD measurements, and analysed the data. S.R.K. performed all density functional theory calculations, supervised by A.W. and D.O.S. M.R. and L.M.H. conducted and analysed OPTP measurements, and contributed to discussions. M.R., T.U. and I.L. performed the PYS measurements, and analysed the results. S.J.Z. performed all the PDS measurements presented in SI, and A.J.S. performed the PDS measurements for Figs. 1c and 4d. S.J.Z. and A.J.S. is supervised by S.D.S. and A.R., respectively. A.J.B. performed ambient-pressure XPS measurements. K.Z. and L.D. performed the

TEM measurements, supervised by L.T.-M. and S.D.S., respectively. J.J. and M.L. performed the ToF-ERDA measurements, which were analysed by M.N. Z.Z. and J.X. contributed to the optimisation of the NC synthesis and purification methods. J.Y. participated in the discussion of physical mechanism. All authors contributed to writing and editing the manuscript.

## Competing interests

The authors declare no competing interests.
