## [Peer Review File · Nature Communications]

REVIEWER COMMENTS

Reviewer #1 (Remarks to the Author):

Referee's comments on the paper "Strong Absorption and Ultrafast Localisation in NaBiS₂ Nanocrystals with Microsecond Charge-Carrier Lifetimes" by Huang et al

This is very interesting paper with several exciting results. In addition to unraveling several interesting physics, the authors point out that the investigated system, ultra-thin films of cation-disordered NaBiS₂ nano-crystals, show highly promising photo-voltaic properties. The work is thorough and the paper is well written although several points have to be elaborated or explained in more detail (see my comments below). I think the work will be of considerable interest to scientists working in the field of photovoltaics and also in thermoelectrics. I recommend publication of the paper after the authors have responded to my comments below.

The main points discussed in this paper are:

The cationic disorder in ABX₂ (where X=S; A=Na,Ag, and B=Bi, and the electronic difference between Na and Ag (which has a filled d shell hybridizing with S p bands) lead to dramatically different photo-physical properties between the two.

AgBiS₂ with cationic disorder has already been shown to show good PV properties. NaBiS₂ in which the S p bands are "pristine" show even better PV response. There are two fundamental differences: the valence S-p band is narrow and rises much faster near the valence band maximum and unusual Na rich environment of S gives rise to highly localized S-p defect states (LDS) above the VBM. These LDS act as deep defect states trapping holes created in the valence band during photo-excitation. Several steady state and pump-probe studies have been made to unravel the complex charge carrier kinetics.

Kelvin probe and photoelectron yield spectroscopy measurements to measure the density of states.

Electronic structure calculations using hybrid DFT including spin-orbit interactions (HSE06+SOC) via the Special Quasirandom Structure (SQS) supercell approach are state-of-the art and provide an excellent picture of the underlying physics.

There are several questions and issues I would like the authors to respond before the paper can be accepted for publication.

(i) The peak in the theoretical DOS above the valence band comes from the localized S-p states associated with S atoms surrounded by 5-6 Na atoms. Why these states are not picked up in the experimental DOS.

(ii) Do the authors understand why these localized states do not exist in AgBiS₂? This system should also have S atom surrounded by 5-6 Ag atoms.

(iii) The authors should give more detailed information about their SQS supercell calculations. Please give the size of the supercell and how many atoms does it contain. This should be given in the text.

(iv) What is the Brillouin Zone corresponding to the band structure given in Fig. 2a. I see 3 S-p bands. Does it mean there is only one S atom in the supercell which has 6 nearest neighbor Na atoms. Does it mean S atoms which have 5 or less Na neighbors are all in the valence band. The authors should explain their theoretical calculations in more detail in the supplementary section.

(v) In this system one should have two types of exciton: one where the hole is in the valence band and the electron is in the conduction band. The authors presumably use a Wannier-Mott (WM) model to estimate their binding energy. There should be a second type of exciton where the hole is in the localized S p-state and the electron is in the conduction band. This exciton cannot be described by the simple WM model.

(vi) "The self-trapping (binding) energy of these small hole polarons is calculated to be ~0.2 eV, with the exact value varying with the local environment (i.e., Na/Bi distribution) about the S²⁻ anion." Give some details about the calculation of the self-trapping energy of a localized hole.

(vii) The statement "While the close-packing of NaBiS₂ bestows high structural dimensionality, the effective electronic dimensionality is vastly reduced due to the spectator nature of Na⁺ and nanoscale heterogeneity in the cation distribution, facilitating this behaviour." needs to be modified. This statement is valid only for certain electronic states. Certainly not for the conduction and valence band states.

(viii) I have questions about what happens in post annealing. The system as synthesized has a high degree of cationic disorder. Do the authors assume that there are no defects in these systems. What happens when you anneal? If the system goes to an ordered or semi-ordered cationic structure then the strongly localized S-p gap states should go away. The configuration with S atom surrounded by 6 Na atoms should go away, taking away the gap states. The authors should clearly define what do they mean and understand about defects in the structurally disordered solid.

In summary, I like this work and strongly recommend publication of the paper in some form.

Reviewer #2 (Remarks to the Author):

The authors presented a thorough characterization of the charge-carrier behaviour in NaBiS₂ nanocrystals, showing that, due to the unique electronic structure of this material, it presents an extremely high absorption coefficient (higher than other materials used for thin film application). On the other hand the authors also revealed an ultra-fast localization of the charges, which detrimental effect for PV can probably be mitigated by the strong absorption and long carrier lifetime.

Therefore this work, which presents new and original results, will be of significance in the field of non-toxic, earth-abundant and air-stable novel PV materials. Moreover, the paper highlights the recurrence of certain opto-electronic features in Bi-containing materials, which are currently being studied as possible substitute for Pb in perovskites.

For this reason I think the work is suitable to be published in Nature Communications after only minor revisions.

I want to congratulate with the authors to have constructed an extremely clear and complete work, that permits to follow easily the authors' discussion even for people not so familiar with ultrafast spectroscopy.

I only have few questions for the authors:

1. How did you calculate the absorption coefficient with a drop casted film?

In order to calculate the absorption coefficient from absorption measurement, thickness of the film is needed. A drop casted film is usually very rough, with inhomogeneous thickness through the substrate area (let's think at the coffee-ring effect). How did you solve this problem? Being the paper presenting an unusually high abs. coeff., this thing should be carefully explained

2. The measurements you carried out are performed on NC. Do you expect to see the same results if you were repeating the measurement in bulk material? I think a general comment of this type would benefit the reader which are not familiar with spectroscopy

Reviewer #3 (Remarks to the Author):

In this study, the optoelectronic properties and charge carrier lifetimes in NaBiS₂ nanocrystals were thoroughly explored with transient absorption and optical pump terahertz probe spectroscopies. The experimental spectroscopy of NaBiS₂ nanocrystals were validated with a detailed theoretical

study that provides valuable insights into the exceptional optical properties of NaBiS₂. The main conclusion that the NaBiS₂ optical properties are derived from the heterogeneity of the Na and Bi atoms is a very plausible conclusion and one that can explain the experimental optical properties. While it is possible to arrive at a theoretical structural model that deviates from reality, the authors further support the theoretical density of states prediction via Kelvin probe and photoelectron yield spectroscopy.

Unfortunately, the authors were not able to provide evidence of any change in the cation homogeneity via annealing or other pre or post processing method. The maximum annealing temperature of 150°C was reportedly used for post processing treatment of the nanocrystals, is it possible to anneal the films at higher temperatures? This may help to change the distribution of Na/Bi ions in the bulk. The authors also claim that the sub gap trap states induced by the annealing post processing is mainly a surface effect. Have the authors tried to anneal a film with a larger nanocrystalline size and therefore different bulk/surface ratios to further support this claim?

According to the structural model found by the Special Quasirandom Structure approach, the heterogeneity of Na/Bi atoms plays a large role in the discussion of the optoelectronic properties. Could the authors clarify the degree of heterogeneity? How many S atoms have non-ideal coordination environments? Previous computational studies on these materials find that sulfur atoms prefer to be coordinated by 3 Na and 3 Bi atoms. From the theoretical work the authors have done, is this still the case?

The Time-of-Flight Elastic Recoil experiment seems to have an issue with underestimating the amount of Na which the authors attribute to Na lost under high vacuum. This is reasonable, however, have the authors done any theoretical calculations of defects potentials for Na vacancies? Can the authors provide an explanation why the Na heterogeneity would manifest itself differently in the optoelectronic properties from Na vacancies?

Overall, the work is of high quality and the results are relatively well supported by both experiment and theory. Addressing the above questions in detail should precede final publication.

In this revised paper, we have now addressed all comments raised, as specified in the point-by-point response below.

The comments from the Reviewers are in blue, our replies are in black. All text quoted from the paper are italicised and in Times New Roman font. All changes to the paper are highlighted in yellow.

Yours Sincerely,

Dr. Robert Hoye, on behalf of all authors

Reviewer 1:

Referee's comments on the paper "Strong Absorption and Ultrafast Localisation in NaBiS₂ Nanocrystals with Microsecond Charge-Carrier Lifetimes" by Huang et al

This is very interesting paper with several exciting results. In addition to unraveling several interesting physics, the authors point out that the investigated system, ultra-thin films of cation-disordered NaBiS₂ nano-crystals, show highly promising photo-voltaic properties. The work is thorough and the paper is well written although several points have to be elaborated or explained in more detail (see my comments below). I think the work will be of considerable interest to scientists working in the field of photovoltaics and also in thermoelectrics. I recommend publication of the paper after the authors have responded to my comments below. The main points discussed in this paper are:

The cationic disorder in ABX₂ (where X=S; A=Na,Ag, and B=Bi, and the electronic difference between Na and Ag (which has a filled d shell hybridizing with S p bands) lead to dramatically different photo-physical properties between the two.

AgBiS₂ with cationic disorder has already been shown to show good PV properties. NaBiS₂ in which the S p bands are "pristine" show even better PV response. There are two fundamental differences: the valence S-p band is narrow and rises much faster near the valence band maximum and unusual Na rich environment of S gives rise to highly localized S-p defect states (LDS) above the VBM. These LDS act as deep defect states trapping holes created in the valence band during photo-excitation. Several steady state and pump-probe studies have been made to unravel the complex charge carrier kinetics. Kelvin probe and photoelectron yield spectroscopy measurements to measure the density of states.

Electronic structure calculations using hybrid DFT including spin-orbit interactions (HSE06+SOC) via the Special Quasirandom Structure (SQS) supercell approach are state-of-the art and provide an excellent picture of the underlying physics.

In summary, I like this work and strongly recommend publication of the paper in some form.

We are thankful to the Reviewer for their thorough reading of the paper and strongly positive and supportive comments.

There are several questions and issues I would like the authors to respond before the paper can be accepted for publication.

(i) The peak in the theoretical DOS above the valence band comes from the localized S-p states associated with S atoms surrounded by 5-6 Na atoms. Why these states are not picked up in the experimental DOS.

We thank the Reviewer for this interesting question. We would firstly note that we do indeed observe these localised states in the highly-sensitive absorbance measurements made by photothermal deflection spectroscopy (PDS), as seen in Fig. 1c of the main text (copied below). However, the absorption coefficient of this sub-gap state is approximately 20 times lower than the absorption coefficient at the optical bandgap of 1.4 eV.

The distinct peak of the localised states is not picked up in the experimental DOS measurements by photoemission yield spectroscopy (PYS), which may be for several reasons:

- The concentration of these localised states, whilst high from the viewpoint of defects, are low compared to the states in the main continuum in the valence band. These relatively low concentrations, coupled with the low transition dipole moment of these localised states make them difficult to detect by PYS;
- The signals of these localised states may be masked by other surface states. PYS is a surface-sensitive technique, and the surface states in the samples could be different from the bulk features predicted from calculations.

Fig. 1 ... **c** Absorption coefficient (α) spectrum of the NaBiS_2 NC film compared with other PV absorbers. ... Theoretical and experimental (Exp.) orbital-projected electronic density of states (DOS) for disordered $Fm\bar{3}m$ NaBiS_2 ...

Nevertheless, we still find a good match between experiment and theory in the upper valence band region. There is a smoother DOS near the valence band maximum (VBM) in experiment (*i.e.*, in the -1 to 0 eV energy range), suggesting that these states are indeed picked up, but with a greater broadening/distribution than the theory result here.

A key contributor to the greater broadening witnessed in experiment is likely the greater range of environments of the localised S p states. The theory calculation shown in Fig. 1e is from a representative supercell calculation with only one Na^+ -rich pocket, whereas the specific $\text{Na}^+/\text{Bi}^{3+}$ concentration in the 2nd coordination shell will also affect the exact energy of these states through electrostatic interactions. This will lead to a distribution of energies for localised S p states in the material, corresponding to a broadening of this DOS peak. By performing

these calculations in a range of supercell sizes, from 40 to 400 atoms, we confirmed the same supercell-independent qualitative behaviour (formation of localised S p states), but also witnessed this variation in their exact energy relative to the ‘delocalised valence band’, depending on the surrounding environment of the Na^+ -coordinated sulfur. All the data for these calculations will be made available in our open-access repository accompanying the manuscript in DOI: 10.14469/hpc/10614. The access code is: ezfu-meme

(ii) Do the authors understand why these localized states do not exist in AgBiS_2 ? This system should also have S atom surrounded by 5-6 Ag atoms.

Yes, Ag^+ -rich pockets should be equally as likely in AgBiS_2 as Na^+ -rich pockets in NaBiS_2 . However, Ag^+ -rich pockets in AgBiS_2 do not give rise to the same high-energy localised S p states, due to the differences in the electronic structure of the upper valence bands between both materials (Fig. 1 e and f in the main text, copied below). In AgBiS_2 , the upper valence band arises from strong orbital repulsion between the filled Ag d states and S p orbitals, which gives rise to an anti-bonding type VBM. This means that even when Ag^+ -rich pockets arise, the S p states remain within the valence band, and thus have greater long-range interactions and not emerge from the valence band continuum.

Fig. 1 ...Theoretical and experimental (Exp.) orbital-projected electronic density of states (DOS) for disordered $\text{Fm}\bar{3}m$ **e** NaBiS_2 and **f** AgBiS_2

For NaBiS_2 on the other hand, Na^+ has no filled valence subshell, and thus acts as a spectator ion that does not contribute to the DOS at the band edges. This results in a lower-energy, non-bonding type VBM in NaBiS_2 , with a very weak Bi-S anti-bonding contribution (also present in AgBiS_2 but drowned out by the far stronger Ag d interaction), thus a narrower bandwidth (reflected by the predicted and measured S p peak at -2 eV in Fig. 1e). This weaker interaction and lower energy of the VBM mean that S p dangling bonds at Na^+ -rich pockets are higher in energy than the non-bonding VB continuum owing to the reduced Coulombic attraction with Na^+ compared to Bi^{3+} , which then arise as localised states within the bandgap. A schematic diagram illustrating this behaviour has now been added to the SI to make our explanations clearer:

Supplementary Figure 5. Schematic Molecular Orbital (MO) diagram of the upper valence band (VB) electronic structure in NaBiS₂ (left) and AgBiS₂ (right). Sulfur orbitals with high coordination of A⁺ cations (denoted as “S 3p A⁺-rich”) have higher energy owing to reduced Coulombic potential from the lower-charge cations (compared to Bi³⁺). From this diagram, it is shown that due to the greater Ag 4d – S 3p repulsion, AgBiS₂ has a larger bandwidth in the VB, such that S 3p (Ag⁺-rich) remains within the VB, whereas in NaBiS₂ S 3p (Na⁺-rich) forms a localised state above the VB maximum.

In addition, we also updated the main text on page 14 and page 17:

Another consequence of the weak anti-bonding VBM character in disordered NaBiS₂, however, is the facilitated emergence of localised S p states just above the ‘bulk’ or ‘delocalised VBM’ (Fig. 1e, Fig. 2a and Supplementary Figure 5) at Na⁺-rich pockets (see next section for a detailed discussion)...

By screening through a range of supercells, it was found that localised S p states tend to form in Na⁺-rich regions, where ≥ 5 of the 6 octahedral coordinating neighbours of S²⁻ (Fig 1a inset and Supplementary Figure 5) are Na⁺ (rather than Bi³⁺)...

(iii) The authors should give more detailed information about their SQS supercell calculations. Please give the size of the supercell and how many atoms does it contain. This should be given in the text.

We have updated the Methods section of the main text to clarify further details of the SQS calculations (changes highlighted in yellow):

To simulate the $Fm\bar{3}m$ disordered rocksalt crystal structure of NaBiS₂, the SQS approach was used⁶⁹, whereby supercells are generated with the cation–cation pair correlations optimized to match that of the ideal infinitely-random distribution. The Alloy Theoretic Automated Toolkit⁷⁰ was used to generate SQS supercells via Monte Carlo–simulated annealing⁷¹. Thirty Monte Carlo simulations were performed for each supercell size, with the structure giving the best match (‘objective function’) to a fully-random material chosen for further calculations.

...
Supercell sizes up to 400 atoms ($28 \times 27 \times 13 \text{ \AA}^3$), were calculated to ensure convergence in the energetic, electronic, optical and polaronic properties. The 400-atom supercell was used for the generation of LDOS plots and charge density isosurfaces, while a well-converged 80 atom supercell ($8 \times 17 \times 14 \text{ \AA}^3$)

was used for calculation of the unfolded bandstructure due to a prohibitive computational cost and data storage requirement (> 1 Tb). Electron and hole polaron calculations were performed by adding/removing an electron to/from the SQS simulation supercells, while 'excitonic' supercells were generated by constraining the total spin to give a triplet state – in each case using 80, 160 and 400 atoms to confirm supercell-independence of the results.

All the data for these calculations will be made available in our open-access repository accompanying the manuscript, deposited in DOI: 10.14469/hpc/10614; access code: ezfu-meme

(iv) What is the Brillouin Zone corresponding to the band structure given in Fig. 2a. I see 3 S-p bands. Does it mean there is only one S atom in the supercell which has 6 nearest neighbor Na atoms. Does it mean S atoms which have 5 or less Na neighbors are all in the valence band. The authors should explain their theoretical calculations in more detail in the supplementary section.

Disordered NaBiS₂ has the $Fm\bar{3}m$ rocksalt crystal structure, thus a face-centred cubic lattice. For determining the reciprocal space path for the band structure shown in Fig. 2a of the main text, we use the conventional Bradley and Cracknell formalism¹. The Brillouin Zone and band structure path are shown in this diagram, which has now been added to the supplementary information:

Supplementary Figure 6. First Brillouin Zone (BZ) of an F-centred cubic lattice, to which the disordered NaBiS₂ crystal space group ($Fm\bar{3}m$) belongs. The high-symmetry BZ points and band path included in the electronic band structure (Fig. 2a) are shown in colour. Vectors $b_{1,2,3}$ denote the cell vectors of the Wigner-Seitz primitive cell.

As well as a reference to this figure in the caption of Fig.2:

...in blue. **Corresponding Brillouin Zone path shown in Supplementary Fig. 6. b (Lower)...**

The Reviewer is correct on all fronts about the band structure. In the 80-atom supercell used for this calculation (details in the Methods section as copied in the response to the previous question), there is one S atom coordinated by 6 Na, which gives rise to the three S *p* localised states in the band gap, and S atoms with a lower Na coordination give rise to states within the valence band. For supercells with a maximum S-Na coordination number of 5, these localised states coincide with or lie slightly above the VBM edge, and again represent the electronic states in which holes can be easily trapped based on the hole polaron calculations.

We have modified the “Formation of Localised S p States” section of the main text on page 17 and the caption below Fig. 2a to clarify this (changes highlighted in yellow):

By screening through a range of supercells, it was found that localised S p states tend to form in Na⁺-rich regions, where ≥ 5 of the 6 octahedral coordinating neighbours of S²⁻ (Fig 1a inset and Supplementary Fig. 5) are Na⁺ (rather than Bi³⁺). This behaviour was found to be consistent across various SQS supercells (from 80 to 400 atoms; see Methods), with local fluctuations of high Na⁺ density giving rise to high-energy localised S p states just above the ‘delocalised VBM’.

...

Fig. 2 Electronic structure and localised states in NaBiS₂. **a** Orbital-projected effective electronic band structure of disordered $Fm\bar{3}m$ NaBiS₂ calculated using hybrid DFT including spin-orbit interactions (HSE06+SOC) via the SQS supercell approach. The supercell contains one S²⁻ anion coordinated by 6 Na⁺ cations, which gives rise to the three localised S p states above the delocalised valence band continuum. Sodium contributions (negligible) in red, bismuth in green and sulphur in blue.

(v) In this system one should have two types of exciton: one where the hole is in the valence band and the electron is in the conduction band. The authors presumably use a Wannier-Mott (WM) model to estimate their binding energy. There should be a second type of exciton where the hole is in the localized S p-state and the electron is in the conduction band. This exciton cannot be described by the simple WM model.

The Reviewer is correct on all fronts, we use a Wannier-Mott model to estimate the exciton binding energy from the VBM and CBM effective masses:

To examine the existence of excitonic peaks as an alternative explanation, we fitted the absorption coefficient spectrum in Fig. 1c based on the Elliott model (see details in Supplementary Note 2). As a result, we obtained a small exciton binding energy of 12 meV, which is in good agreement with the calculated value of 27 meV for disordered NaBiS₂ using the Wannier-Mott hydrogenic model.

However, excitons formed from the localised S p state and the conduction band electron cannot be described by the WM model (as the Reviewer has pointed out) because (1) the

effective mass description breaks down for the localised states, and (2) the assumption of bulk-like screening ($\epsilon = \epsilon_\infty$) between the electron-hole pair within the WM model does not hold. In order to model these excitons, we performed spin-constrained DFT calculations in our SQS supercells (again 80 to 400 atoms). These calculations reveal that electron-hole pairs prefer to dissociate in the presence of the locally-heterogeneous electronic potential, due to fluctuations in the Na/Bi atomic densities. Here these fluctuations are strong enough to overcome the electrostatic attraction of hole and electron wavefunctions in the bound-exciton state, so that holes preferentially localise at the Na⁺-rich pockets and electrons in the Bi³⁺-rich regions, as shown in Fig. 4c:

Fig. 4 ... c Charge density isosurfaces of a relaxed electron-hole pair in a disordered NaBiS₂ supercell with sodium, bismuth, and sulphur atoms in pink, blue and yellow respectively. The transparent yellow and blue surfaces represent electron and hole densities respectively

This behaviour can also be understood from the viewpoint of the electronic structure. In Na⁺-rich pockets, the highest energy occupied states are the localised S *p* states where holes can be easily trapped. While in Bi³⁺-rich pockets, lower energy conduction band states will form owing to greater hybridisation as well as stronger electrostatic attraction, and electrons will thus tend to localise in these regions. Therefore, we do not report “exciton binding energies” for these cases, as the total energies of these supercells are approximately equal to the sum of the isolated hole and electron polaron binding energies, as expected. We have modified the discussion on page 29 in the main text to clarify these results:

*On the other hand, our calculations reveal that electron polarons are only weakly-localised in Bi³⁺-rich regions (Fig. 4c), but this localisation process can be strengthened by a heterogeneous electronic potential due to atomic-scale fluctuations in the cation distribution (giving local Na⁺-rich and Bi³⁺-rich regions). Our supercell calculations of electron-hole pairs in disordered NaBiS₂ reveal spontaneous separation of the excited carriers into isolated polarons, rather than the formation of bound excitons. Thus, whilst excitons can form between the CBM and continua in the VB, we do not expect excitons to form between the CBM and localised S *p* states. Furthermore, the This localisation of electron and hole polarons at Bi³⁺-rich and Na⁺-rich pockets, respectively, results in a spatial segregation of the excited electrons and holes (Fig. 4c), which likely inhibits the formation of self-trapped excitons, and accounts for the absence of photoluminescence in NaBiS₂.*

Details of the polaron and electron-hole pair calculations are then given in the Methods section.

(vi) “The self-trapping (binding) energy of these small hole polarons is calculated to be ~0.2 eV, with the exact value varying with the local environment (i.e., Na/Bi distribution) about the S2- anion.” Give some details about the calculation of the self-trapping energy of a localized hole.

We have added the following clarifications to the Methods section regarding the calculation of the polaron trapping/binding energies:

Electron and hole polaron calculations were performed by adding/removing an electron to/from the SQS simulation supercells, while ‘excitonic’ supercells were generated by constraining the total spin to give a triplet state – in each case using 80, 160 and 400 atoms to confirm supercell-independence of the results. A range of initial perturbations to the initial structure were tested using the Bond Distortion Method², to aid polaron trapping, though in each case the unperturbed structure also relaxed to the localised self-trapped state (indicating negligible trapping barriers). Polaron trapping/binding energies are taken from total energy differences of the hole or electron-containing supercells before and after relaxation of the atomic coordinates.

This is the conventional approach for calculating small polaron binding energies in computational simulations of solid-state materials, for example see Ref. 2 at the end of this response document.

(vii) The statement “While the close-packing of NaBiS₂ bestows high structural dimensionality, the effective electronic dimensionality is vastly reduced due to the spectator nature of Na⁺ and nanoscale heterogeneity in the cation distribution, facilitating this behaviour.” needs to be modified. This statement is valid only for certain electronic states. Certainly not for the conduction and valence band states.

We thank the Reviewer for pointing this out. Indeed, this statement should only refer to the band-edge states but not the conduction or valence band continua. We have modified the statement accordingly on page 27:

While the close-packing of NaBiS₂ bestows high structural dimensionality, the effective electronic dimensionality of the band-edge states is vastly reduced due to the spectator nature of Na⁺ and nanoscale heterogeneity in cation distribution. This leads to localised 0D S p states in the highest-occupied (hole) electron bands (Na⁺-rich pockets), and the lowest-energy unoccupied (electron) states primarily inhabiting Bi³⁺-rich regions in the material.

To note, as mentioned above, we also find the lowest-energy conduction band minimum states to exhibit reduced electronic dimensionality, since electrons have strong preference towards the regions with even just a slightly higher Bi³⁺ concentration. This result has been confirmed in different supercells, as displayed in Figure R1:

400-Atom Disordered Supercell:

160-Atom Disordered Supercell: 80-Atom Disordered Supercell:

Figure R1. Charge density isosurfaces of a relaxed electron-hole pair in disordered NaBiS_2 supercells with 400, 160 and 80 atoms, respectively. In these figures, the sodium, bismuth, and sulfur atoms are depicted in pink, blue and yellow respectively. The transparent yellow surfaces represent the electron densities. This data is included in our open-access repository accompanying the manuscript, which will be uploaded to DOI: 10.14469/hpc/10614. Access code: ezfu-meme

(viii) I have questions about what happens in post annealing. The system as synthesized has a high degree of cationic disorder. Do the authors assume that there are no defects in these systems. What happens when you anneal? If the system goes to an ordered or semi-ordered cationic structure then the strongly localized S-p gap states should go away. The configuration with S atom surrounded by 6 Na atoms should go away, taking away the gap states. The authors should clearly define what do they mean and understand about defects in the structurally disordered solid.

Our response to this question has two parts: 1) what we understand by defects in NaBiS_2 , and 2) whether a homogeneous cation distribution could be achieved through post-annealing.

What we understand by defects in NaBiS_2 :

We do not assume no defects in the as-synthesized nanocrystals, but we do think that more surface defects are introduced into the nanocrystals after post-annealing. Point defects can still occur in a cation-disordered solid because the NCs are crystalline with well-defined cation and anion sites in a rocksalt structure. Therefore, cation and anion vacancies can occur, as can interstitials, or anti-sites between cation and anion sites (e.g., Na_S or S on a cation site). However, we note that the formation energy of cation vacancies depends on the local (in)homogeneity in the cation distribution, which changes with the level of cation disorder. Considering the case for Na vacancies, we find that their formation energies do vary in different distances away from localised S p states. This result has also been included in the Supplementary Fig. 20 (copied below) in the SI.

Supplementary Figure 20. Formation energies of neutral (V_{Na}^0) and negatively-charged (V_{Na}^{-1}) Na vacancies in disordered (SQS) $Fm\bar{3}m$ NaBiS_2 , as a function of vacancy site distance from the localised

S p state (see Fig. 2 in main text), using Γ -only k -point sampling and Na-poor chemical conditions. Formation energies for the negatively-charged vacancies (V_{Na}^{-1}) correspond to a Fermi level positioned at the (0/-1) charge transition level of the lowest energy vacancy (i.e. where $E_F(V_{\text{Na}}^0) = E_F(V_{\text{Na}}^{-1})$).

We also find that the sub-bandgap absorption of NaBiS₂ will increase after post-annealing at different temperatures (in an inert environment), as shown in Supplementary Fig. 10b (copied below). We believe that this increase in sub-bandgap absorption is due to the introduction of sub-gap defect states, which could arise from the removal of organic ligands from the surface of NCs or the fusion of NCs.

Supplementary Figure 10. ... **b** Absorption coefficient of pristine and annealed NaBiS₂ films at different temperatures....

To clarify what we mean by defects in a cation-disordered solid, and also expand upon the discussion around the defect introduction during post-annealing, we have made the following changes (highlighted) in the “Influence of defects” section of the main text on pp. 33 and 34:

Whilst NaBiS₂ is a cation-disordered solid, these NCs are crystalline with well-defined cation and anion sites in a rocksalt crystal structure. Thus, point defects (e.g., cation/anion vacancies, interstitials, cation on anion and anion on cation anti-sites) can still occur. Although large defect concentrations are very unlikely to form at such low annealing temperatures, oleic acid or oleylamine ligands surrounding the NaBiS₂ NCs could be detached during heating and remove surface species such as Bi atoms, which may create dangling bonds and thus defect states on the NC surface.

...

Also, annealed NaBiS₂ films showed a more inhomogeneous morphology with several voids, which indicates that NCs could have fused together after ligands removal, as shown in Supplementary Fig. 14. This NC fusion process can also introduce sub-gap trap states by forming ‘necked dimers’, as previously found in PbS quantum dots⁶¹.

Cation homogeneity through post-annealing:

We do not expect NaBiS₂ to transform into an ordered / semi-ordered structure after post-annealing because (1) a disordered rocksalt cubic structure remains in the sample annealed at up to 150 °C (see Supplementary Fig. 10a, copied below), albeit with slight peak shifts, and

(2) a more disordered structure should be favoured at higher temperatures from the perspective of thermodynamics. Nevertheless, the small XRD peak shifts in the annealed samples suggest that “cation homogeneity” might be slightly improved so that cation-anion lengths are decreased (Supplementary Fig. 11, copied below). If the cation distribution macroscopically over the lattice has indeed become more homogeneous (but which would increase disorder on the short scale), we would expect an elimination of the localised S p states and enhanced absorption in the above-gap region, as shown in the case of annealed AgBiS₂³. However, from Supplementary Fig. 10b (copied above), we can see that this is not the case. This result indicates even though cation homogeneity can be improved in annealed NaBiS₂, the degree to which it is improved is not as significant as that previously reported in annealed AgBiS₂, which can be ascribed to the different orbital compositions near the band edges of both materials. We have modified the discussion in the “Influences of defects” section of the main text (changes highlighted in yellow) on page 32, as well as to the SI, to further clarify our conclusions:

We found that all post-annealed films remained in the cation-disordered rocksalt phase. But in the case of films annealed 150 °C, we observed a slight shift in XRD peaks to higher diffraction angles (Supplementary Fig. 10a). The reduced lattice constants associated with these peak shifts may be due improved cation homogeneity, which is predicted to lead to reduced cation-anion bond lengths (Supplementary Fig. 11), consistent with previous reports of post-annealed AgBiS₂¹⁴. But unlike AgBiS₂, we found post-annealed NaBiS₂ to exhibit decreased absorbance in the above-gap region and increased absorbance in the sub-gap region (Supplementary Fig. 10b). Further increasing the annealing temperature of NaBiS₂ to higher values results in the degradation of the material to orthorhombic Bi₂S₃ from 250 °C (Supplementary Fig. 12). Thus, over the limited range of post-annealing temperatures available, NaBiS₂ remains in the disordered phase, and we are limited in the extent to which cation homogeneity could be improved, such that localised states cannot be eliminated.

Supplementary Figure 10. ... a XRD patterns of pristine and annealed NaBiS₂ films at 150 °C. The peak positions for the pristine NaBiS₂ film are displayed as dash-lines for comparison.

Supplementary Figure 11. Average calculated **a** Na-S and **b** Bi-S bond lengths for disordered ($Fm\bar{3}m$) $NaBiS_2$ in a representative 400-atom SQS supercell relaxed using hybrid DFT, as a function of distance from the Bi-rich region of the supercell. In agreement with observations for $AgBiS_2$, regions of greater cation homogeneity (far from the Bi-rich region) show decreased cation-anion bond lengths.

Reviewer 2:

The authors presented a thorough characterization of the charge-carrier behaviour in $NaBiS_2$ nanocrystals, showing that, due to the unique electronic structure of this material, it presents an extremely high absorption coefficient (higher than other materials used for thin film application). On the other hand the authors also revealed an ultra-fast localization of the charges, which detrimental effect for PV can probably be mitigated by the strong absorption and long carrier lifetime.

Therefore this work, which presents new and original results, will be of significance in the field of non-toxic, earth-abundant and air-stable novel PV materials. Moreover, the paper highlights the recurrence of certain opto-electronic features in Bi-containing materials, which are currently being studied as possible substitute for Pb in perovskites.

For this reason I think the work is suitable to be published in Nature Communications after only minor revisions.

I want to congratulate with the authors to have constructed an extremely clear and complete work, that permits to follow easily the authors' discussion even for people not so familiar with ultrafast spectroscopy.

We thank the Reviewer for their strongly positive comments, and their appraisal of the significance of the work for the community, as well as the clarity and thorough nature of the paper we have written.

I only have few questions for the authors:

1. How did you calculate the absorption coefficient with a drop casted film? In order to calculate the absorption coefficient from absorption measurement, thickness of the film is needed. A drop casted film is usually very rough, with inhomogeneous thickness through the substrate area (let's think at the coffee-ring effect). How did you solve this problem? Being the paper presenting an unusually high abs. coeff., this thing should be carefully explained

We calculated the absorption coefficient α by using the formula below:

$$\alpha = \frac{\ln\left(\frac{1-R}{T}\right)}{d}$$

where d is the film thickness, and R and T are the reflectance and transmittance, respectively, determined from ultraviolet-visible spectrophotometry (UV-Vis) equipped with an integrating sphere. Indeed, as pointed out by the Reviewer, we need to know the film thickness d in order to calculate the absorption coefficient α . Here, we used atomic force microscopy (AFM) to sweep across a step-edge made in the same drop-cast film measured in UV-Vis. To overcome the inhomogeneity issue of drop-cast films, we found that drying the NCs deposited from a solution in hexane in an argon-filled glovebox could result in homogeneous films with no coffee-ring patterns. The image of a drop-cast film prepared in an argon-filled glovebox has been shown in Fig 1b (copied below). As can be seen, there are no coffee-ring patterns in our samples.

Figure 1 ... **b** XRD patterns and photographs of NaBiS₂ NC films synthesised at 150 °C on the same day of preparation (Day 0) and after 112 days (Day 112) of storage in ambient air (60-70% relative humidity).

In addition, a typical cross-sectional profile of the step-edge made in the drop-cast film prepared in an argon-filled glovebox is shown below and added into SI now:

Supplementary Figure 3. Profile of the step-edge from the substrate to drop-cast NaBiS₂ film (dried in an Ar-filled glovebox) measured by atomic force microscopy (AFM). The film thickness is homogeneous with a thickness variation less than 10 nm. The peak at the step edge was from the accumulated NCs when using a blade to scratch the film in order to create a step-edge.

We now have clarified the details above to the “Synthesis, Stability and Absorption Characteristics of NaBiS₂ Nanocrystals” section in the main text on page 11 and Methods section:

To examine the absorption features in detail, we used ultraviolet-visible spectrophotometry (UV-Vis) to determine the absolute absorption coefficients (α) and photothermal deflection spectroscopy (PDS) to resolve the absorption profile more accurately at the band edge and below (details in Methods). Homogeneous films without any coffee-ring-like patterns (Fig. 1b, inset and Supplementary Fig. 3) were achieved by drop-casting the NC solution onto a glass substrate in an Ar-filled glovebox.

Methods

...

Absorption measurements. The absorption coefficient α is calculated from equation 2 below:

$$\alpha = \frac{\ln\left(\frac{1-R}{T}\right)}{d} \quad (2)$$

where R and T are the reflectance and transmittance, respectively, of NaBiS₂ drop-cast films, and d is the film thickness. R and T were measured by UV-Vis within an integrating sphere, and d was determined from the cross-sectional profile of the step-edge across from a substrate to a film using an atomic force microscope (Nanoscope III), as shown in Supplementary Fig. 3. To ensure homogeneous films were prepared for these measurements, we drop-cast the NC solution onto a glass substrate in an Ar-filled glovebox. The hexane solvent used for the NC solution evaporated rapidly, and it can be seen from Fig. 1b that this resulted in uniform films with no coffee-ring patterns. Atomic force microscopy measurements showed that variations in film thickness were only on the order of 10 nm (Supplementary Fig. 3). The relative absorbance spectrum measured by PDS was normalised to its highest signal value. We then mapped this maximum value of the absorbance, found at 3.1 eV photon energy, onto the absolute absorption coefficient value obtained from UV-Vis at the same photon energy. The whole absolute absorption coefficient spectrum could then be acquired.

2. The measurements you carried out are performed on NC. Do you expect to see the same results if you were repeating the measurement in bulk material? I think a general comment of this type would benefit the reader which are not familiar with spectroscopy

Indeed, we would expect to see similar results in bulk disordered NaBiS₂. All the calculations pertain to the bulk properties of this material. For experiments, the key properties measured are: 1) the high absorption coefficient and strong absorption onset, 2) photoconductivity transients, and 3) recombination rate of charge-carriers. We will examine each of these three key properties in detail below.

For the high absorption coefficient: this was determined by UV-vis, which is a bulk measurement technique. Our computational analysis shows that this arises due to the band structure of the material (as explained in the section title “Understanding the strong absorption of NaBiS₂ through first-principles calculations” of the main text), and is influenced by cation homogeneity. We would therefore expect to see the same effects in bulk NaBiS₂.

For the decay in photoconductivity after photoexcitation: this is measured by optical pump terahertz probe (OPTP) spectroscopy. These transients are dominated by the most mobile charge-carriers, which will be the charge-carriers within the bulk of each NC (especially the electrons, which are less localised than holes). This is because the NCs are coordinated by long-chain organic ligands, which make inter-NC transport difficult. We note that in the OPTP measurements, the NC film is photoexcited with a pulsed laser of 400 nm wavelength. Owing to the high absorption coefficient, the absorption depth is around 20 nm, and the majority of the signal will come from the top 20 nm of the sample. However, each NC is 5±1 nm in size, so we are still obtaining signal from a cluster of many NCs.

Finally, for the measurements of the recombination rate by transient absorption spectroscopy: a similar argument applies because a 400 nm wavelength pulsed excitation is also used in the short-time transient absorption measurements. Thus, the recombination rate measured will be averaged over $\sim 10^{11}$ NCs from a volume of $\sim 5 \times 10^{12}$ nm³. The slow decay in ground state bleach in the long-time transient absorption measurements is also fully consistent with carrier localisation (which is an intrinsic bulk property of the material) and we would expect this effect to occur in bulk materials.

Therefore, these key measurements are representative of the bulk properties of NaBiS₂. That being said, it is possible that in changing to a bulk thin film or single crystal, we may observe different localisation or relaxation rates that arise from reduced surface area-to-volume ratios (as surface interactions are known to favour cation segregation in these systems) and different synthesis conditions. We are also open to the possibility of geometric confinement effects from the NCs, as recently observed in perovskite NCs⁴. We have added the following comment to the end of the 'Mechanism for Charge-Carrier Localisation' section on page 31:

We note that similar results would be expected in the case of bulk disordered NaBiS₂ films, though likely with differences in the localisation rates, due to the potentially weaker geometric confinement,⁵⁹ as well as the effects of greatly-reduced surface area, or different synthesis conditions on the degree of cation heterogeneity.

Reviewer 3:

In this study, the optoelectronic properties and charge carrier lifetimes in NaBiS₂ nanocrystals were thoroughly explored with transient absorption and optical pump terahertz probe spectroscopies. The experimental spectroscopy of NaBiS₂ nanocrystals were validated with a detailed theoretical study that provides valuable insights into the exceptional optical properties of NaBiS₂. The main conclusion that the NaBiS₂ optical properties are derived from the heterogeneity of the Na and Bi atoms is a very plausible conclusion and one that can explain the experimental optical properties. While it is possible to arrive at a theoretical structural model that deviates from reality, the authors further support the theoretical density of states prediction via Kelvin probe and photoelectron yield spectroscopy.

We thank the Reviewer for their careful evaluation of our paper, and their interest in our work. The questions raised below are very interesting, and we have addressed them individually.

Unfortunately, the authors were not able to provide evidence of any change in the cation homogeneity via annealing or other pre or post processing method. The maximum annealing temperature of 150C was reportedly used for post processing treatment of the nanocrystals, is it possible to anneal the films at higher temperatures? This may help to change the distribution of Na/Bi ions in the bulk.

We thank the Reviewer for this suggestion and have accordingly annealed the NaBiS₂ films at higher temperatures. From the XRD patterns shown in Figure R2 below, we observed that the diffraction peaks shift towards larger angles after post-annealing at 150 °C and 200 °C for 1 hour compared to the pristine NCs without post-annealing. The smaller lattice parameters may arise from a reduction in the cation-S bond lengths, which is consistent with a more homogeneous cation distribution, as found in our calculations of NaBiS₂ (Supplementary Fig. 11, copied below) and previously observed for AgBiS₂³. Interestingly, unlike the case with post-annealed AgBiS₂ films, we still did not observe a significant trend of peak narrowing in annealed NaBiS₂ films, which indicates that the crystallinity was not improved in these samples.

Figure R2. The XRD patterns of NaBiS₂ NC films annealed at different temperatures

Supplementary Figure 11. Average calculated **a** Na-S and **b** Bi-S bond lengths for disordered ($Fm\bar{3}m$) NaBiS₂ in a representative 400-atom SQS supercell relaxed using hybrid DFT, as a function of distance from the Bi-rich region of the supercell. In agreement with observations for AgBiS₂², regions of greater cation homogeneity (far from the Bi-rich region) show decreased cation-anion bond lengths.

However, even with the XRD evidence suggesting improved cation homogeneity, we still saw a decrease in the absorption coefficient in the above-bandgap region and increased absorption in the sub-gap region for these annealed NaBiS₂ films, as shown in Figure R3 below. These trends are the same as the samples annealed below 150 °C, and are opposite to the trends previously reported for AgBiS₂³. The increase in sub-gap absorption with increased annealing may arise due to the introduction of defect states, such as through NC fusion or the removal of surface ligands (now elaborated on further in the main text on page 33).

For near-bandgap absorption, the different trend for NaBiS₂ than AgBiS₂ may be due to the different orbital composition at the band edges. In AgBiS₂, the band edges are comprised of different states: Bi p – S p for the CBM, and Ag d – S p for the VBM. In this case, cation inhomogeneity will result in reduced absorption due to the spatial separation of the band-extrema. But in NaBiS₂, both band-extrema are derived from Bi-S interactions (Bi p – S p for CBM, S p with some Bi s for VBM). The absorption coefficient in NaBiS₂ is therefore expected

to be less sensitive to cation inhomogeneity. Therefore, we believe that even though post-annealing treatment at higher temperatures may improve the homogeneity of cations in disordered NaBiS₂, it does not enhance the absorption strength in the same way as previously found for AgBiS₂.

Figure R3. The absorption coefficient spectrum of NaBiS₂ NC films annealed at different temperatures.

Further increasing the annealing temperature of NaBiS₂ up to 250 °C and 310 °C resulted in a gradual transition from rocksalt NaBiS₂ to orthorhombic Bi₂S₃, as shown from **a** to **e** in Supplementary Figure 11, copied below. This shows that there is a limit in the post-annealing temperature range for NaBiS₂. Therefore, other processing (e.g., ligand-exchange) or synthesis methods will be needed for effectively tuning cation homogeneity in this material.

We have added these results, and interwoven a discussion incorporating them, as well as the points discussed above into the “Influence of defects” section of the main text on page 32, as well as to the SI:

We found that all post-annealed films remained in the cation-disordered rocksalt phase. But in the case of films annealed 150 °C, we observed a slight shift in XRD peaks to higher diffraction angles (Supplementary Fig. 10a). The reduced lattice constants associated with these peak shifts may be due improved cation homogeneity, which is predicted to lead to reduced cation-anion bond lengths (Supplementary Fig. 11), consistent with previous reports of post-annealed AgBiS₂¹⁴. But unlike AgBiS₂, we found post-annealed NaBiS₂ to exhibit decreased absorbance in the above-gap region and increased absorbance in the sub-gap region (Supplementary Fig. 10b). Further increasing the annealing temperature of NaBiS₂ to higher values resulted in the degradation of the material to orthorhombic Bi₂S₃ from 250 °C (Supplementary Fig. 12). Thus, over the limited range of post-annealing temperatures available, NaBiS₂ remains in the disordered phase, and we are limited in the extent to which cation homogeneity could be improved, such that localised states cannot be eliminated.

Supplementary Figure 10. ... **a** XRD patterns of pristine and annealed NaBiS_2 films at $150\text{ }^\circ\text{C}$. The peak positions for the pristine NaBiS_2 film are displayed as dash-lines for comparison.

Supplementary Figure 12. XRD patterns of NaBiS_2 films post-annealed at temperatures from a $150\text{ }^\circ\text{C}$ up to e $310\text{ }^\circ\text{C}$ for 1 hour in Ar -filled glovebox. The rocksalt cubic phase of NaBiS_2 ($Fm\bar{3}m$) starts to transform into orthorhombic Bi_2S_3 ($Pbnm$) after annealing at $250\text{ }^\circ\text{C}$ or higher. The reference pattern for disordered rocksalt NaBiS_2 (ICSD data base, Coll. Code: 616841)1 and Bi_2S_3 (ICSD data base, Coll. Code: 89324)5 are displayed in **a** and **e**, respectively.

The authors also claim that the sub gap trap states induced by the annealing post processing is mainly a surface effect. Have the authors tried to anneal a film with a larger nanocrystalline size and therefore different bulk/surface ratios to further support this claim?

We thank the Reviewer for the suggestion and have investigated the post-annealing effect on larger NCs of NaBiS₂. If defects are more likely to be introduced on the surface of the NCs, we will expect large NCs to exhibit less significant changes in sub-gap absorption compared to smaller NCs. Here, we compared the (normalised) sub-gap absorption of small (mean size ~ 5 nm, determined from TEM images) and large NCs (mean size ~ 18 nm, determined from TEM images) annealed at 150 °C for 1 hour in an argon-filled glovebox. As expected, we do see a smaller change in the sub-gap absorption of larger NCs, which was only enhanced by less than an order of magnitude (~ 8 times). On the other hand, small NCs exhibited a ~ 30 times enhancement in the sub-gap absorption. We have now added this result to the “Influence of defects” section of the main text on page 33 and SI (changes highlighted in yellow):

Although large defect concentrations are very unlikely to form at such low annealing temperatures, oleic acid or oleylamine ligands surrounding the NaBiS₂ NCs could be detached during heating and remove surface species such as Bi atoms, which may create dangling bonds and thus defect states on the NC surface. To verify this postulation, we compared the change in sub-gap absorption of films composed of small NCs (mean size ~ 5 nm, determined from TEM images) versus films composed of large NCs (mean size ~ 18 nm, determined from TEM images) after the same post-annealing treatment. If defects are mainly introduced to the NC surface after annealing, we would expect to observe a smaller change in sub-gap absorption for larger NCs owing to their lower surface area-to-volume ratio. We indeed found this to be the case, as shown in Supplementary Fig. 13.

Supplementary Figure 13. Photothermal deflection spectroscopy (PDS) absorbance spectrum of NaBiS₂ films composed of **a** large NCs (mean size ~ 18 nm) and **b** small NCs (mean size ~ 5 nm). These spectra are normalised to the absorbance at 3.1 eV. The spectrum of the pristine film and film annealed at 150 °C for 1 hour in an argon-filled glovebox are displayed in each sub-Figure.

According to the structural model found by the Special Quasirandom Structure approach, the heterogeneity of Na/Bi atoms plays a large role in the discussion of the optoelectronic properties. Could the authors clarify the degree of heterogeneity? How many S atoms have non-ideal coordination environments? Previous computational studies on these materials find that sulfur atoms prefer to be coordinated by 3 Na and 3 Bi atoms. From the theoretical work the authors have done, is this still the case?

The degree of heterogeneity in the synthesised NaBiS₂ NCs depends closely on the synthesis and treatment conditions. Previous works have found ligand interactions to strongly favour cation segregation in ABZ₂ NCs^{5,6}, with a strong affinity for one of the cation species at the surface, and this is likely one of the driving factors of cation heterogeneity. Our theoretical results reveal a low thermodynamic cost of heterogeneity in the cation distribution, in addition to the fact that this could be kinetically stabilised during the growth of NCs in solution, indicating a high prevalence of non-ideal coordination environments for S atoms in the synthesised material.

Our results agree with previous computational studies^{7,8}, which found that the $R\bar{3}m$ trigonal ordered structure is the thermodynamic ground state, with S atoms coordinated by 3 Na and 3 Bi. However, our results also reveal the low enthalpy cost of site disorder on the NaBiS₂ lattice, which in combination with ligand interactions yields the disordered rocksalt structure commonly reported in the literature^{5,7,9,10}. The ideal coordination of S in the disordered structure is still 3 Na and 3 Bi, however the random placement of Na and Bi atoms on the disordered cation sublattice results in a Gaussian distribution (with a mean coordination of 3 Na and 3 Bi) for the S coordinations, with no strong disfavouring of non-ideal environments. When local fluctuations in the cation distribution occur, which yields Na⁺-rich pockets (corresponding to the tail region of the S coordination distribution, with 5 or 6 Na⁺ neighbours), these fast-trapping localised S *p* states arise. Similar formation of non-ideal environments and localised states in the presence of cation disorder have been noted in related materials (A^{II}B^{IV}N₂ and kesterites). We have added the following to the 'Formation of Localised S *p* States' section in the main text:

*This behaviour was found to be consistent across various SQS supercells (from 80 to 400 atoms; see Methods), with local fluctuations of high Na⁺ density giving rise to high-energy localised S *p* states just above the 'delocalised VBM'. We note that similar formation of localised anion *p* states at regions of low electronic potential, namely clusters of low-valence (A^{III}) cations, have recently been reported in the related A_{II}B_{IV}N₂ disordered compounds (including MgSnN₂, ZnSnN₂, ZnGeN₂, and others)⁴⁵⁻⁴⁷ as well as disordered kesterites (CZTS)⁴⁸.*

The Time-of-Flight Elastic Recoil experiment seems to have an issue with underestimating the amount of Na which the authors attribute to Na lost under high vacuum. This is reasonable, however, have the authors done any theoretical calculations of defects potentials for Na vacancies? Can the authors provide an explanation why the Na heterogeneity would manifest itself differently in the optoelectronic properties from Na vacancies?

Na vacancies (or other defects) could also potentially introduce sub-gap states which manifest in the optical absorption spectrum. However, this would require the presence of an optical transition for the defect at the energy corresponding to the sub-gap peak measured experimentally, in addition to a high concentration (on the order of 1 in ~20-5000 unit cells, depending on the transition dipole moment, judging from the relative intensities of the sub-gap and band-edge absorbance). Both of these conditions are satisfied by the localised S *p* states with Na heterogeneity, while for defects, a concentration this high is unlikely.

To confirm this hypothesis, we have performed calculations of Na vacancies (V_{Na}) in ordered and disordered NaBiS₂. The results have been added to the SI, and discussed in detail in Supplementary Note 5, along with Supplementary Figures 18-23. The key finding is that Na vacancies have a small but significant formation energy as expected (~0.5 eV for a Fermi level located near mid-gap, under Na-poor conditions). This agrees with the hypothesis that Na could be readily lost under high vacuum, but that V_{Na} should not be initially present in the high concentrations required to match the intensity of the sub-gap peak measured. For example, under Na-poor conditions (most favourable for Na vacancies to form), with a Fermi level at 0.4

eV above the VBM, the equilibrium concentration of V_{Na} at room temperature in as-synthesised disordered NaBiS_2 is calculated to be ~ 1 in 10^7 unit cells.

Moreover, when no localised S p states (due to Na heterogeneity) are present (as in ordered NaBiS_2), Na vacancies are found to behave as shallow/resonant acceptors, and so are not expected to contribute strongly to ultra-fast carrier trapping, nor to have a defect level which would align with the position of the sub-gap peak in the experimental absorption.

The transition level only becomes deep when the Na vacancies are present in the Na^+ -rich pockets with localised S p states. However, these deep states can only form in a small subspace (Na^+ -rich pockets). The density of these Na^+ -rich pockets is already low. Added to this is the fact that these trap states have higher formation energies than elsewhere in the material. We would therefore expect the concentration of these deep traps to be to very small, and insufficient to yield the fast carrier trapping witnessed experimentally. Furthermore, the (0/-1) transition levels are so deep that they are closer to the CBM than the VBM (see Supplementary Fig. 22), and this is another factor suggesting that these states would not facilitate the fast localisation of holes. Finally, the relative energy of the transition levels to the VBM (>0.6 eV) also does not match with the energy of the sub-bandgap peak compared to the optical bandgap (<0.4 eV). We therefore conclude that if deep Na vacancies form in the presence of localised S p states, they would still not account for the sub-gap absorption peak or carrier localisation in NaBiS_2 .

In addition to adding the defect calculation results to the SI, we also added the following discussion to the 'Mechanism of Carrier Localisation' in the main text on page 29:

We note that Na vacancies were also investigated as a potential origin of the sub-gap absorption and fast trapping in NaBiS_2 (details in Supplementary Note 5). However, we found them to be shallow acceptors. The exception was when these vacancies were located in Na^+ -rich pockets, which then gave rise to deep traps. However, the concentration of these deep states would be too low to account for fast carrier trapping because of their higher formation energies than elsewhere in the material, and the fact that they can only form in the Na^+ -rich pockets, which already have low concentrations. Furthermore, the (0/-1) transition levels are so deep they are closer to the CBM than VBM, which would also not facilitate fast hole trapping, and is inconsistent with the energy of the sub-gap absorption peak relative to the optical bandgap found experimentally (Fig. 1c).

Overall, the work is of high quality and the results are relatively well supported by both experiment and theory. Addressing the above questions in detail should precede final publication.

We would like to thank the Reviewer again for their time, and for their appraisal that our work is of high quality. We hope the changes we made to the paper, and new data taken satisfactorily address the questions raised.

References for the response letter:

1. Bradley, C. & Cracknell, A. *The mathematical theory of symmetry in solids: representation theory for point groups and space groups*. (Oxford University Press, 2010).
2. Pham, T. D. & Deskins, N. A. Efficient Method for Modeling Polarons Using Electronic Structure Methods. *J. Chem. Theory Comput.* **16**, 5264–5278 (2020).
3. Wang, Y. *et al.* Cation disorder engineering yields AgBiS_2 nanocrystals with enhanced optical absorption for efficient ultrathin solar cells. *Nat. Photonics* **16**, 235–241 (2022).
4. Hempel, H. *et al.* Predicting Solar Cell Performance from Terahertz and Microwave Spectroscopy. *Adv. Energy Mater.* **12**, 2102776 (2022).

5. Medina-Gonzalez, A. M., Rosales, B. A., Hamdeh, U. H., Panthani, M. G. & Vela, J. Surface Chemistry of Ternary Nanocrystals: Engineering the Deposition of Conductive NaBiS₂ Films. *Chem. Mater.* **32**, 6085–6096 (2020).
6. Viñes, F., Konstantatos, G. & Illas, F. Matildite Contact with Media: First-Principles Study of AgBiS₂ Surfaces and Nanoparticle Morphology. *J. Phys. Chem. B* **122**, 521–526 (2018).
7. Syam Kumar, R., Akande, A., El-Mellouhi, F., Park, H. & Sanvito, S. Theoretical investigation of the structural, elastic, electronic, and dielectric properties of alkali-metal-based bismuth ternary chalcogenides. *Phys. Rev. Mater.* **4**, 075401 (2020).
8. Gabrel'yan, B.V., Lavrentiev, A. A., Nikiforov, I. Y. & Sobolev, V.V. Electronic energy structure of MBiS₂ (M = Li, Na, K) calculated with allowance for the difference between the M-S and Bi-S bond lengths. *J. Struct. Chem.* **49**, 788–794 (2008).
9. Glemser, O. & Filcek, M. Über Alkalithioismutate (III). *Zeitschrift für Anorg. und Allg. Chemie* **279**, 321–323 (1955).
10. Boon, J. W. The crystal structure of NaBiS₂ and KBiS₂. *Recl. des Trav. Chim. des Pays-Bas* **63**, 32–34 (1944).

Policy checklist

We have now completed the editorial policy checklist, and this is uploaded with our revised submission.

Raw data openly available for the paper

We have uploaded all of our raw data (experimental and computational) onto DOI: 10.14469/hpc/10614, and have added the following data availability statement to the paper:

Data Availability

The experimental and computational data generated in this paper and in the Supplementary Information have been deposited to the open access database under accession code <http://dx.doi.org/10.14469/hpc/10614>

Please note that for now we have embargoed the deposited data. The access code is: ezfu-meme, and we welcome the Editor and Reviewers to access this data.

REVIEWERS' COMMENTS

Reviewer #1 (Remarks to the Author):

The authors have gone through the comments of all the reviewers carefully. Their response is quite detail and appears to be correct and proper. The response to my comments is quite satisfactory. I think the paper has improved considerably after the authors' modifications. I recommend publication of the paper without further review.

Reviewer #2 (Remarks to the Author):

I thank you the authors for the answers. I am satisfied with the modifications and explanations provided. I think the paper is ready to be accepted.

Reviewer #3 (Remarks to the Author):

I think the authors have done a good job at addressing the questions posed by the other reviewers and I. At this stage, I would recommend accepting this work.

Reviewer comments:

Reviewer #1: The authors have gone through the comments of all the reviewers carefully. Their response is quite detail and appears to be correct and proper. The response to my comments is quite satisfactory. I think the paper has improved considerably after the authors' modifications. I recommend publication of the paper without further review.

Reviewer #2

(Remarks to the Author): I thank you the authors for the answers. I am satisfied with the modifications and explanations provided. I think the paper is ready to be accepted.

Reviewer #3

(Remarks to the Author): I think the authors have done a good job at addressing the questions posed by the other reviewers and I. At this stage, I would recommend accepting this work.

Response: We are delighted that all three Reviewers consider our revised paper to satisfactorily address all points previously raised, and would like to thank them all again for their time and thorough evaluation of our work.